# Defects in t$^6$A tRNA modification due to *GON7* and *YRDC* mutations lead to Galloway-Mowat syndrome

Christelle Arrondel et al.[#]

N$^6$-threonyl-carbamoylation of adenosine 37 of ANN-type tRNAs (t$^6$A) is a universal modification essential for translational accuracy and efficiency. The t$^6$A pathway uses two sequentially acting enzymes, YRDC and OSGEP, the latter being a subunit of the multiprotein KEOPS complex. We recently identified mutations in genes encoding four out of the five KEOPS subunits in children with Galloway-Mowat syndrome (GAMOS), a clinically heterogeneous autosomal recessive disease characterized by early-onset steroid-resistant nephrotic syndrome and microcephaly. Here we show that mutations in *YRDC* cause an extremely severe form of GAMOS whereas mutations in *GON7*, encoding the fifth KEOPS subunit, lead to a milder form of the disease. The crystal structure of the GON7/LAGE3/OSGEP subcomplex shows that the intrinsically disordered GON7 protein becomes partially structured upon binding to LAGE3. The structure and cellular characterization of GON7 suggest its involvement in the cellular stability and quaternary arrangement of the KEOPS complex.

---

Transfer RNAs (tRNA) are subject to multiple post-transcriptional modifications that are important for the stabilization of their ternary structure and the precision of the decoding process[1]. The majority of the complex modifications are concentrated in the anticodon region of the tRNAs and are crucial for accuracy of protein synthesis. The threonylcarbamoylation of the $N^6$ nitrogen of the adenosine at position 37 ($t^6A$) of most ANN-accepting tRNAs represents one of the very few nucleotide modifications that exists in every domain of life[2,3]. The $t^6A$ biosynthesis pathway consists of two steps: firstly, the YRDC enzyme (Sua5 in yeast) synthesizes an unstable threonylcarbamoyl-AMP intermediate (TC-AMP) and in a second step, the KEOPS protein complex transfers the TC-moiety from TC-AMP onto the tRNA substrate[4]. Enzymes that synthesize TC-AMP exist in two versions depending on the organism: a short form which only has the YrdC domain (such as human YRDC) and a long form which has an extra Sua5 domain (yeast SUA5 for instance). The eukaryotic KEOPS complex contains five subunits GON7/LAGE3/OSGEP/TP53RK/TPRKB that are arranged linearly in that order[5–8]. OSGEP is the catalytic subunit that carries out the TC-transfer reaction and its orthologs are present in virtually all sequenced genomes[9]. The other subunits are essential for the $t^6A$ modification of tRNA, but their precise roles are as yet unknown, especially that of GON7, an intrinsically disordered protein (IDP), which was only recently identified in humans[5,7]. Fungal Gon7 was shown to be an IDP that adopts a well-defined structure covering 50% of its sequence upon complex formation with Pcc1 (LAGE3 homolog)[8]. In humans, GON7 was recently shown to be also structurally disordered in absence of the other KEOPS complex subunits. GON7 was proposed to be a very remote homolog of the yeast Gon7 protein although its structure upon complex formation remains unknown[5,7].

tRNA modifications have been demonstrated to play a role in the development of the brain and nervous system, and an increasing number of defects in these modifications are now being linked to various human neurodevelopmental disorders[10]. We recently identified autosomal recessive mutations in genes encoding four of the five subunits of human KEOPS complex, namely *LAGE3*, *OSGEP*, *TP53RK,* and *TPRKB* in patients with Galloway-Mowat syndrome (GAMOS, OMIM#251300). GAMOS is a rare neuro-renal disorder characterized by the co-occurrence of steroid-resistant nephrotic syndrome (SRNS) with microcephaly and neurological impairment[11]. GAMOS is clinically heterogeneous, reflecting a genetic heterogeneity. Indeed, disease-causing mutations have been identified in eight genes to date: four in KEOPS genes and four in other unrelated genes, *WDR73*, *WDR4*, *NUP133*, and *NUP107* (refs. [12–17]). SRNS is typically detected in the first months of life and most often rapidly progresses to end-stage renal disease (ESRD) within a few months; however, there are rarer cases with preserved renal function in adulthood[18]. Kidney lesions range from minimal change disease, to focal segmental glomerulosclerosis (FSGS) that might be of the collapsing type, or diffuse mesangial sclerosis (DMS). Cerebral imaging findings include cerebral and cerebellar atrophy, and gyration and/or myelination defects. These anomalies are associated with neurological deficits such as psychomotor impairment, hypotonia, seizures, and more rarely sensorineural blindness and deafness. Affected children may also present with facial and/or skeletal dysmorphic features. The prognosis of GAMOS is poor, and most affected children die before 6 years of age.

Here we present 14 GAMOS-affected individuals from seven families, with mutations in *GON7* (alias *C14orf142*) and *YRDC*, both genes encoding proteins involved in the biosynthesis of the $t^6A$ modification. These data, together with our previous work, show that mutations in genes encoding all the proteins involved in the two chemical steps of $t^6A$ lead to GAMOS. Furthermore, we determine the crystal structure of the GON7/LAGE3/OSGEP KEOPS subcomplex showing that GON7 becomes structured upon binding to LAGE3. The structure also explains our observations that GON7 stabilizes the remainder of the KEOPS complex and directs its quaternary organization.

## Results

**Identification of *GON7* and *YRDC* mutations in GAMOS patients**. Through whole-exome sequencing in individuals with GAMOS, we identified mutations in the *GON7* gene in 11 affected individuals from 5 unrelated families and in the *YRDC* gene in 3 affected individuals from 2 unrelated families (Fig. 1a–d and Supplementary Table 1). Four of the families with *GON7* mutations (Families A to D) carried the same homozygous nonsense mutation (c.21 C>A, p.Tyr7*) which causes a stop codon at position 7 of the protein. These families, all originating from the same region of Algeria, shared a common haplotype at the *GON7* locus indicating a founder effect (Supplementary Table 2). The affected individual of the fifth family (Family E) carried a different mutation involving the same residue at position 7 and leading to a frameshift (c.19dup, p.Tyr7Leufs*16). Both *GON7* mutations are predicted to lead to the absence of protein expression and, as expected, no protein was detected in cells available from the affected individuals from family A, B, and C (Supplementary Fig. 1a, c). Two compound heterozygous *YRDC* mutations were identified in Family F: a missense mutation (c.251 C>T, p.Ala84Val) and a 4-base pair deletion leading to a frameshift (c.721_724del, p.Val241Ilefs*72). In Family G, we identified a homozygous in-frame deletion of Leucine 265 (c.794_796 del, p.Leu265del). For both families, western blot and qPCR analysis on cell extracts from affected children showed the presence of YRDC transcripts and proteins (Supplementary Fig. 1b, d). To make a prediction of the effect of the *YRDC* mutations on the protein, we created a three-dimensional (3D) structural model of human YRDC using the structure of the YRDC domain of the archaeal Sua5 (PDB 4E1B, 20% sequence identity[19]) and mapped these mutations onto this in silico model (Fig. 1e). Structures of YRDC domains are very well conserved and sequence alignment shows that the human YRDC only has minor insertions/deletions compared to Sua5 (Supplementary Fig. 2). The replacement of Ala84, located in a hydrophobic region between a β-sheet and a connecting α-helix, by the larger amino acid valine might perturb optimal packing and destabilize the structure of the protein. The YRDC Leu265del mutation affects a highly conserved amino acid and creates a deletion in a C-terminal peptide that hangs over the active site and could have a role in enzyme activity.

Early-onset proteinuria was observed in all affected children, with first detection ranging from between birth and 5 years. All but three children reached ESRD between 1.5 months and 6 years of age. All individuals carrying *YRDC* mutations presented with congenital or infantile SRNS detected from between birth and 4 months of age and died early, whereas most of the individuals carrying *GON7* mutations were alive at last follow-up, with either a functioning graft or with normal renal function despite a mild to heavy proteinuria (Supplementary Table 1). Kidney biopsies, when available, typically displayed FSGS (Families A, C, and E) or DMS (Families B, F, and G) (Fig. 2a–d). In addition to developmental delay, primary microcephaly was present in the two affected children of one family with *YRDC* mutations, whereas the affected child in the second *YRDC* family and all *GON7*-mutated individuals presented with post-natal microcephaly. Brain magnetic resonance imaging revealed a spectrum of cerebellar and cortical hypoplasia or atrophy with thin corpus callosum and ventricular dilation, myelination delay, and in one case a simplified gyral pattern (Individual G.II-2) (Fig. 2e–n,

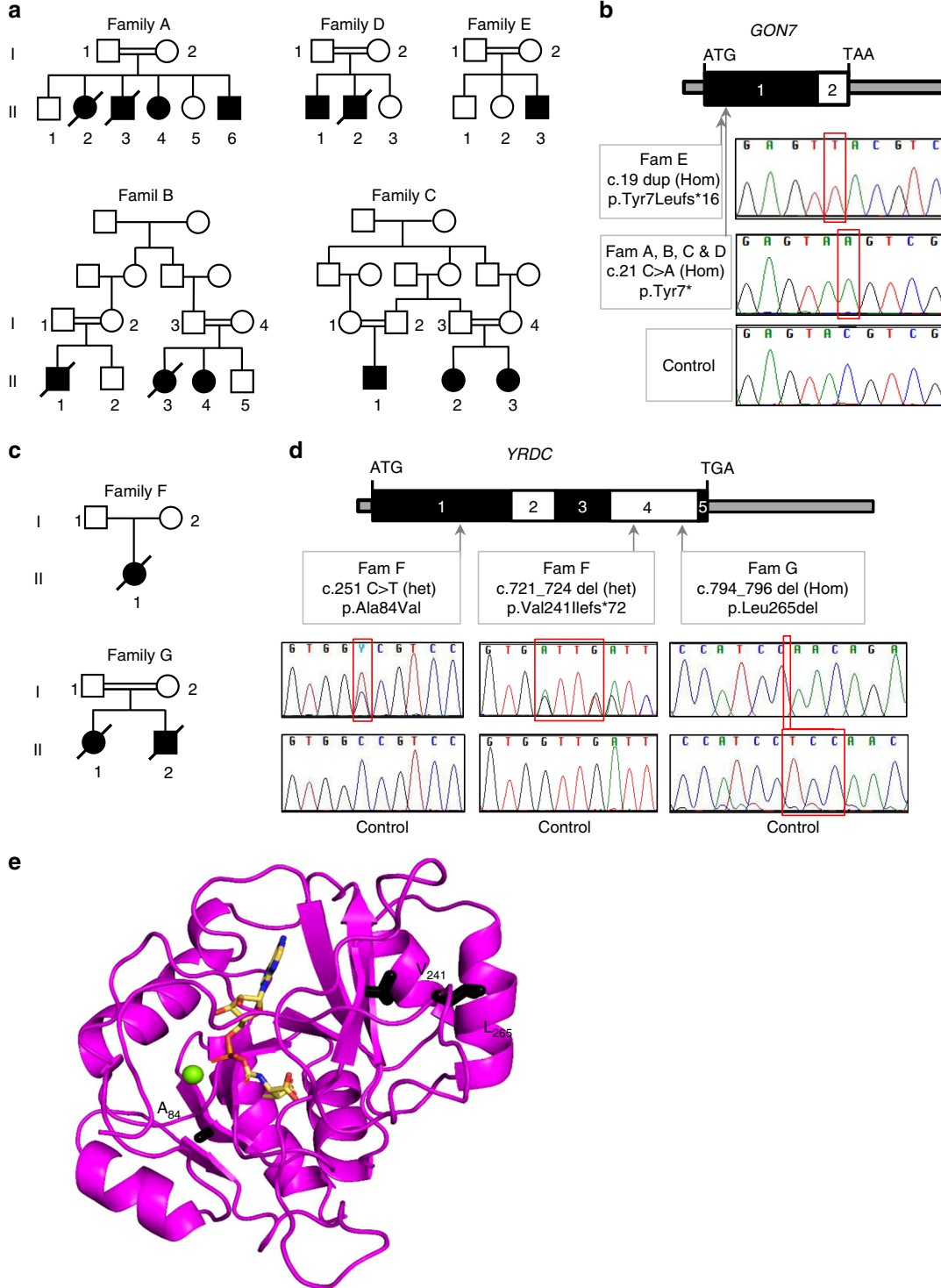

**Fig. 1** Identification of mutations in *GON7* and *YRDC* in patients with Galloway-Mowat syndrome. **a**, **c** Pedigrees of families with mutations in *GON7* (**a**) and in *YRDC* (**c**). Affected individuals are in black. **b**, **d** Organization of exons of human *GON7* and *YRDC* cDNAs. Positions of start and stop codons are indicated. Arrows indicate positions of the identified mutations. Lower panels show the sequencing traces for affected individuals with identified mutated nucleotide indicated with a red square (Hom: homozygous; het, heterozygous). **e** Representation of a 3D model of human YRDC, bound to the reaction product threonylcarbamoyl-adenylate, in sticks. The model was constructed using the crystal structure of *Sulfolobus tokodaii* Sua5 (PDB code 4E1B). The side chains of the three mutated residues are in black sticks. The green sphere represents an $Mg^{2+}$ ion

Supplementary Fig. 3). Extra-renal features included facial dysmorphy, arachnodactyly, hiatal hernia with gastro-esophageal reflux, congenital hypothyroidism (solely in the *YRDC* cases), and myoclonia. This clinical picture is highly reminiscent of that observed in GAMOS-affected individuals with mutations in *LAGE3, OSGEP, TP53RK,* and *TPRKB*[13]. However, individuals with *GON7* mutations presented with milder neurological and renal manifestations, always with post-natal microcephaly and no gyration defects, later onset of proteinuria (median age 18 months vs. 1) and slower progression to ESRD

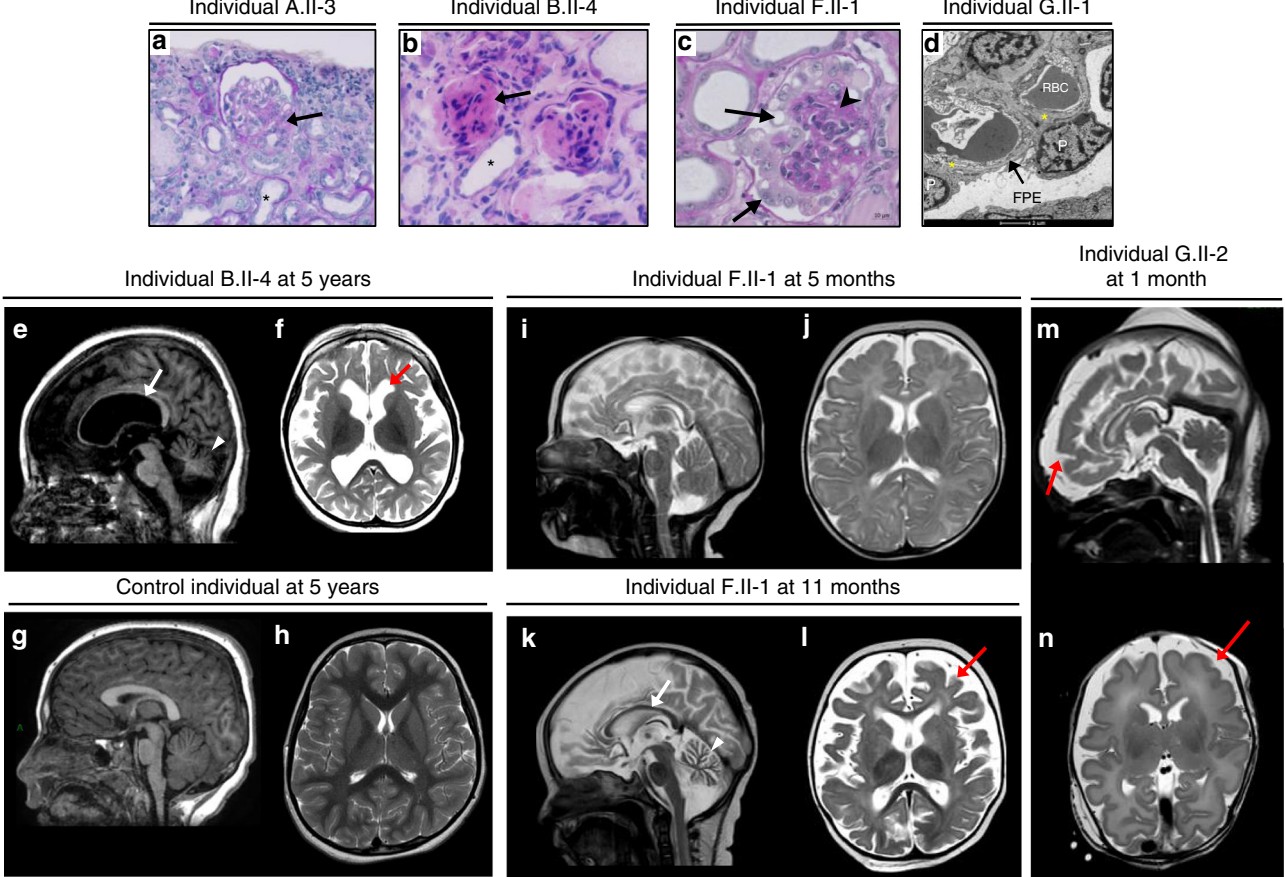

**Fig. 2** Kidney pathology analysis and neuroimaging. Light and transmission electron microscopy (TEM) of kidney sections of patients with *GON7* (**a**, **b**) or *YRDC* mutations (**c**, **d**). **a** Individual A.II-3 displays a retracted glomerulus with a focal segmental glomerulosclerosis lesion at the vascular pole (black arrow) and tubular dilations (black star) (PAS; ×200 magnification). **b** Individual B.II-4 displays diffuse mesangial sclerosis with tiny, retracted and sclerosed glomeruli (black arrow) with dilated tubes surrounded by flat epithelial cells (black star) and interstitial fibrosis (H&E; ×400 magnification). **c** Individual F.II-1 displays a marked glomerular tuft collapsing (arrowhead) surrounded by a layer of enlarged and vacuolized podocytes (black arrows) (PAS stain; ×400 magnification, scale bar, 10 μm). **d** TEM of individual G.II-1 shows diffuse foot process effacement (FPE; black arrow), a classical hallmark of nephrotic syndrome, along a glomerular basement membrane (GBM) with abnormal folded and laminated segments (yellow stars), alternating with others with normal appearance. P podocyte, RBC red blood cell. Scale bar, 2 μm. Brain MRI of patients with *GON7* (**e**, **f**) and *YRDC* mutations (**i–n**). **e**, **f** Brain MRI abnormalities in individual B.II-4 at 5 years. Sagittal T1-weighted image (**e**) shows important cortical subtentorial atrophy as well as corpus callosal (arrow) and cerebellar atrophy (arrowhead). The axial T2-weighted image (**f**) shows abnormal myelination and ventricular dilatation (red arrow). **g**, **h** Brain MRI of a 5-year old control showing sagittal T1 (**g**) and axial T2 (**h**) weighted images. **i–l** Brain MRI abnormalities in individual F.II-1 at 5 months (**i**, **j**) and 11 months (**k**, **l**). Sagittal T2-weighted image shows normal pattern at 5 months (**i**) evolving to a progressive major cerebellar (arrowhead) and cortical atrophy with a very thin corpus callosum (arrow) at 11 months (**k**). The axial T2-weighted image is normal at 5 months (**j**) but shows a very marked abnormality of myelination and cortical atrophy (red arrow) at 11 months (**l**). **m**, **n** Brain MRI abnormalities in individual G.II-2 at 1 month. Sagittal T2 (**m**) and axial T2 (**n**) weighted images show gyral anomalies with marked frontal gyral simplification (red arrow) and myelination delay

(median age 49 months in 8/11 children vs. 5 months in 3/3 children), and a longer survival compared to the *YRDC* cases.

**Impact of *YRDC* and *GON7* mutations on t⁶A biosynthesis.** To assess the pathogenicity of *YRDC* mutations, we first used a yeast heterologous expression and complementation assay as previously performed for *OSGEP* mutations identified in GAMOS individuals[13]. Indeed, the deletion of *SUA5*, the *YRDC* ortholog in *S. cerevisiae* leads to a very severe growth defect, similarly to the deletion of each of the genes encoding the five KEOPS subunits[3,6,20,21]. We therefore heterologously expressed the human YRDC cDNAs encoding wild-type (WT) and mutant proteins in the *Δsua5* strain to evaluate their ability to rescue the slow growth phenotype. Since the catalytic activity of YRDC does not require protein partners, the WT YRDC protein efficiently complemented the *Δsua5* growth defect (Fig. 3a). Although a

somewhat similar complementation was observed for the p. Ala84Val and p.Leu265del mutants, the p.Val241Ilefs*72 mutant was notably unable to improve the poor growth of the *Δsua5* strain (Fig. 3a). All YRDC proteins were efficiently expressed in *Δsua5* strain, except the p.Val241Ilefs*72 mutant that was barely detectable by western blot, suggesting that it is likely being degraded by an intracellular proteolytic machinery (Fig. 3b). Using mass spectrometry, we then analyzed the t⁶A content of these transformed *Δsua5* yeast strains. As expected, since Sua5 is the only enzyme in yeast that generates TC-AMP, the *Δsua5* mutant was unable to synthetize t⁶A, whereas the WT YRDC expressing strain exhibited t⁶A levels comparable with those measured for WT Sua5 (Fig. 3c). The p.Ala84Val and p.Leu265del mutants showed a slight, but significant decrease in t⁶A levels. In contrast, like in the *Δsua5* strain, no trace of t⁶A modification could be detected in the p.Val241Ilefs*72 mutant (Fig. 3c). In line with the results of the growth complementation assay, there was a

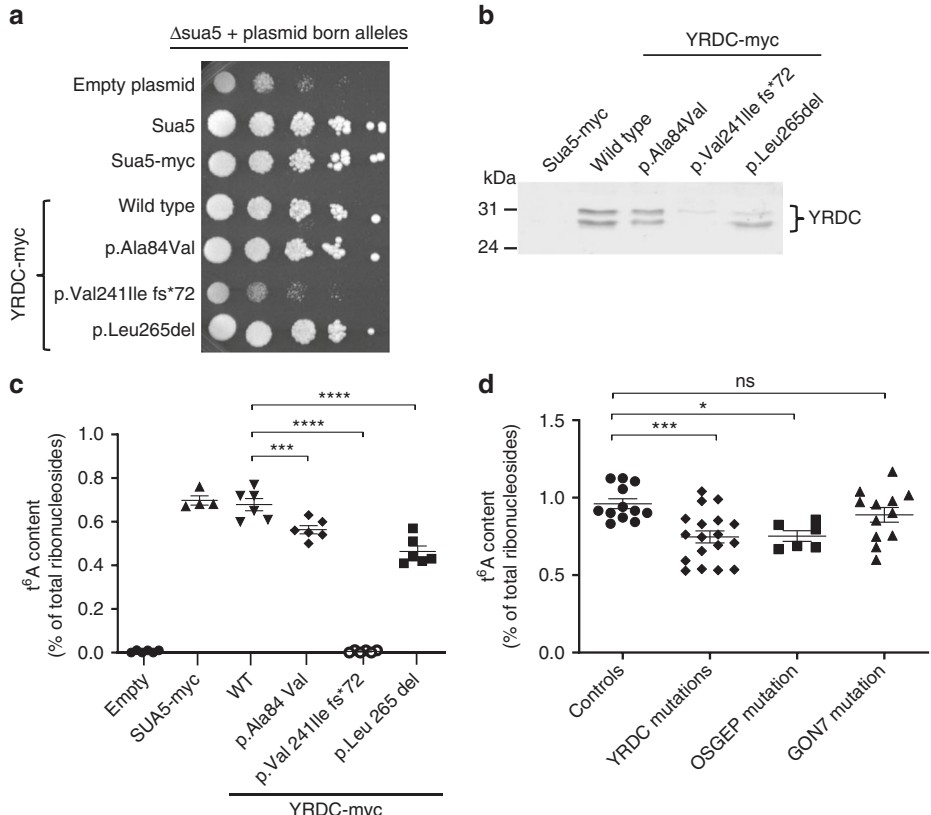

**Fig. 3** Effects of *YRDC* and *GON7* mutations on t⁶A biosynthesis. **a** Evaluation of fitness of Δ*sua5* yeast strains expressing human YRDC variants (spots are from 10-fold serial dilutions of cell suspensions at $OD_{600nm} = 0.5$, and three independent clones were evaluated) and **b** western blot analysis on total protein extracts from Δ*sua5* yeast cells expressing human *YRDC* variants using anti-hYRDC antibody. **c, d** Mass spectrometry (LC-MS/MS) quantification of t⁶A modification in total tRNAs extracted from Δ*sua5* yeast cells expressing human YRDC variants (**c**) (mean ± s.e.m. of two independent LC-MS/MS experiments (technical replicates), each measuring samples from three independent yeast transformants; one-way ANOVA ($F$ (5,28) = 269.4, $p <$ 0.0001), Dunnett's multiple comparisons test, ***$p = 0.0008$, ****$p < 0.0001$) and from cultured primary skin fibroblasts from controls (two unaffected individuals) and affected individuals with either the p.Tyr7* *GON7* mutation (two individuals), or the *YRDC* mutations (three individuals) or with the p. Arg352Gln *OSGEP* mutation in the homozygous state (individual «CP» described in Braun et al.[13]) (**d**) (mean ± s.e.m. of two independent LC-MS/MS experiments (technical replicates), each measuring samples from three independent cell culture experiments; one-way ANOVA ($F$ (3,44) = 6.446, $p <$ 0.001), Dunnett's multiple comparisons test, n.s. = 0.4894, *$p = 0.0169$, ***$p = 0.008$). Source data are provided as a Source Data file

direct correlation between cell fitness and t⁶A content. This allowed us to classify *YRDC* mutations into hypomorphic (encoding p.Ala84Val and p.Leu265del) and amorphic (encoding p.Val241Ilefs*72) alleles, as has been previously shown for *OSGEP* mutations[13]. A similar approach could not be applied for *GON7* mutations since GON7 failed to complement the growth defect of the Δ*gon7* yeast strain (see ref. [7] and our data). We therefore measured the t⁶A content in fibroblasts from two individuals with the p.Tyr7* *GON7* mutation, three individuals with *YRDC* mutations and one individual with the p.Arg325Gln *OSGEP* mutation. The t⁶A levels were significantly decreased in both *YRDC*- and *OSGEP*-mutated fibroblasts, and to a lesser extent in *GON7*-mutated fibroblasts (Fig. 3d) confirming the impact of these mutations on t⁶A biosynthesis in affected individuals. In addition, we demonstrated that, similarly to individuals with *OSGEP* or *TP53RK* mutations[13], telomere length was not affected in individuals with *YRDC* and *GON7* mutations (Supplementary Fig. 4). This confirms that contrary to what has been demonstrated in yeast, human YRDC, and KEOPS complex are not involved in telomere maintenance in human cells[22–24].

**In vitro characterization of WT and mutants YRDC**. To compare the stability and structure of the WT YRDC with those of the p.Ala84Val and p.Leu265del mutants, we expressed and purified these proteins in an *E. coli* expression system (Supplementary Table 3). The three proteins could be purified but we noticed that both mutants were less stable and less soluble compared to the WT (Supplementary Methods). To probe the proper folding of the YRDC WT and mutants, we collect 1D ¹H-NMR spectra (Supplementary Fig. 5a). All the spectra displayed well-dispersed signals for amide protons as well as several signals at chemical shifts lower than 0.8 ppm that are typical of methyl groups in the hydrophobic core of proteins, suggesting the WT and mutants were well folded. To compare their enzymatic properties, we measured their TC-AMP synthesizing activities in vitro by quantifying the pyrophosphate reaction product. The p.Ala84Val and p.Leu265del mutants have lost about 75% and 30% of their activities respectively compared to WT (Supplementary Fig. 5b). The activities of these mutants are compatible with their hypomorphic nature, as suggested by the results of the yeast Δ*sua5* complementation experiments (Fig. 3a).

**Proliferation, apoptosis, and protein synthesis defects**. We have previously shown that transient gene expression knockdown (KD) of human KEOPS components *OSGEP*, *TP53RK*, and *TPRKB* leads to perturbations of various cellular processes including proliferation and apoptosis[13]. Similarly here, we transiently depleted the expression of *GON7* and *YRDC*, as well as *LAGE3* and *OSGEP* as

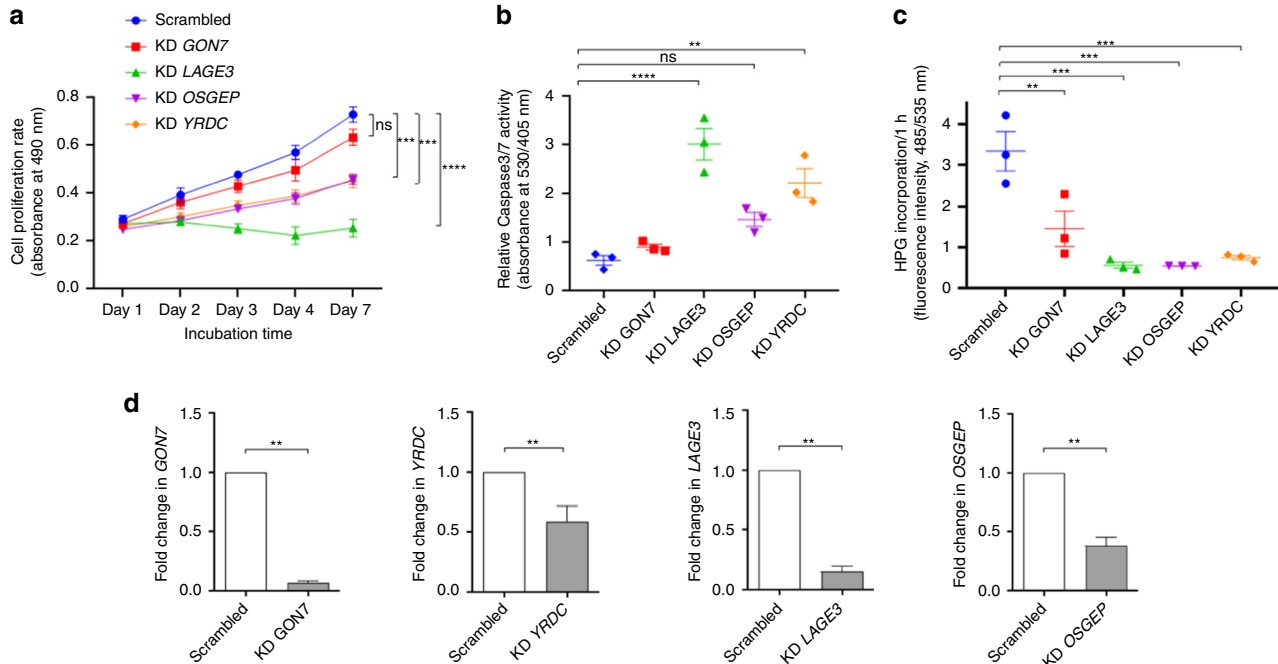

**Fig. 4** Proliferation, apoptosis, and protein synthesis defects upon *GON7* and *YRDC* knockdown. Transient knockdown (KD) of *GON7*, *LAGE3*, *YRDC*, and *OSGEP* was performed by lentiviral transduction of shRNA in immortalized human podocyte cell lines with a scrambled (non-targeting) shRNA as control. **a** Cell proliferation was assessed using a colorimetric MTT assay over 7 days, measuring absorbance at 490 nm at days 1, 2, 3, 4, and 7 (mean ± s.e.m. of $n = 5$ experiments, with each experiment performed in triplicate; two-way ANOVA ($p < 0.0001$), Dunnett's multiple comparisons test, n.s. = 0.2031, ***$p < 0.0007$, ****$p < 0.0001$). **b** Cell apoptosis was evaluated by quantification of caspase 3/7 activation on the basis of fluorescence intensity (530/405 nm). Absolute values were normalized to DAPI fluorescence intensity as an internal control and compared to non-targeting shRNA-treated control cells (scrambled) (mean ± s.e.m. of $n = 3$ experiments with each experiment performed in triplicate; one-way ANOVA ($F (4,10) = 21.42$, $p < 0.0001$), Dunnett's multiple comparisons test, n.s. = 0.0556, **$p = 0.0012$, ****$p < 0.0001$). **c** Protein biosynthesis rates were assessed on the basis of incorporation of HPG, an alkyne-containing methionine analog. After 2 h, alkyne-containing proteins were quantified on the basis of fluorescence intensity (485/535 nm). Absolute values were normalized to DAPI fluorescence intensity as an internal control and compared to control cells (mean ± s.e.m. of $n = 3$ experiments, with each experiment performed in triplicate, one-way ANOVA ($F (4,10) = 16.36$, $p = 0.0002$), Dunnett's multiple comparisons test, **$p = 0.0035$, ****$p < 0.0003$). **d** Relative expression of *GON7*, *YRDC*, *LAGE3*, and *OSGEP* transcripts were normalized to that of HPRT in KD podocytes compared to non-targeting shRNA control treated cells (mean ± s.e.m. of $n = 5$ experiments, with each experiment being performed in triplicate; two-tailed Mann–Whitney test, **$p < 0.05$). Source data are provided as a Source Data file

positive controls, in an immortalized human podocyte cell line[25]. We then demonstrated using a colorimetric cell proliferation assay that diminished expression of all four of these genes decreased cell proliferation, with the strongest decrease being observed in *LAGE3* KD podocytes (Fig. 4a). Despite efficient *GON7* KD, cells exhibited only a slight decrease in cell proliferation compared to cells treated with the control scrambled shRNA (Fig. 4a, d). The impairment of cell proliferation in *YRDC* and *OSGEP* KD cells was less marked than in *LAGE3* KD cells, which could be explained by a less efficient gene silencing (Fig. 4d). By measuring Caspase-3/7 activity, we next demonstrated that apoptosis was inversely related to proliferation with *LAGE3* KD podocytes displaying the highest rate of apoptosis (Fig. 4b). Since loss of t⁶A modification impacts global translation in yeast[26], we also quantified the newly synthetized protein levels, which were decreased in all KD cells (Fig. 4c), even in *GON7* KD podocytes where proliferation and apoptosis rates were not drastically affected (Fig. 4a, b). Altogether, these results reinforce our previous findings for the other KEOPS subunits, OSGEP, TP53RK, and TPRBK, and confirm that mutations which alter t⁶A biosynthesis in human cells have an impact on cell survival through decreased proliferation and protein synthesis, ultimately leading to apoptosis.

**Structure of the human GON7/LAGE3/OSGEP subcomplex**. To better understand the role of human GON7 and how its loss of function could be connected with GAMOS, we set out to determine its structure and to establish how it interacts with the other KEOPS subunits. We had either crystal structures (TPRKB) or good quality 3D models (LAGE3, OSGEP, TP53RK) for all of the KEOPS subunits at our disposal, except for GON7 (ref. [6]). Based on very weak sequence similarity, it was proposed, that GON7 is a remote homolog of yeast Gon7 (ref. [7]). We first investigated the structure of GON7 in solution by collecting a 2D ¹H-¹⁵N Band-Selective Optimized Flip Angle Short Transient Heteronuclear Multiple-Quantum Correlation (SOFAST-HMQC) NMR spectrum of a ¹⁵N-labeled GON7 sample. The poor spectral dispersion in the ¹H dimension of the 2D correlation spectrum showed that GON7 lacks well-defined structure, confirming the conclusions of Wan et al.[7]. Adding non-labeled LAGE3 to the sample, caused the shift and/or disappearance for many crosspeaks, suggesting GON7 interacts with LAGE3 (Supplementary Fig. 6a). We further characterized the conformation of GON7 in solution by small-angle X-ray scattering (SAXS) measurements. By representing the scattering data as a dimensionless Kratky plot ($qR_g^2 \times I_q/I_0$ versus $qR_g$), one can assess qualitative information on the degree of compactness of the scattering object[27]. The plateau observed for GON7 is characteristic of a fully disordered protein, possibly with very short elements of secondary structure (Fig. 5a), confirming our NMR data (Supplementary Fig. 6a). We then purified the recombinant GON7/LAGE3/OSGEP complex and analyzed its behavior in solution by SAXS. Our SAXS data

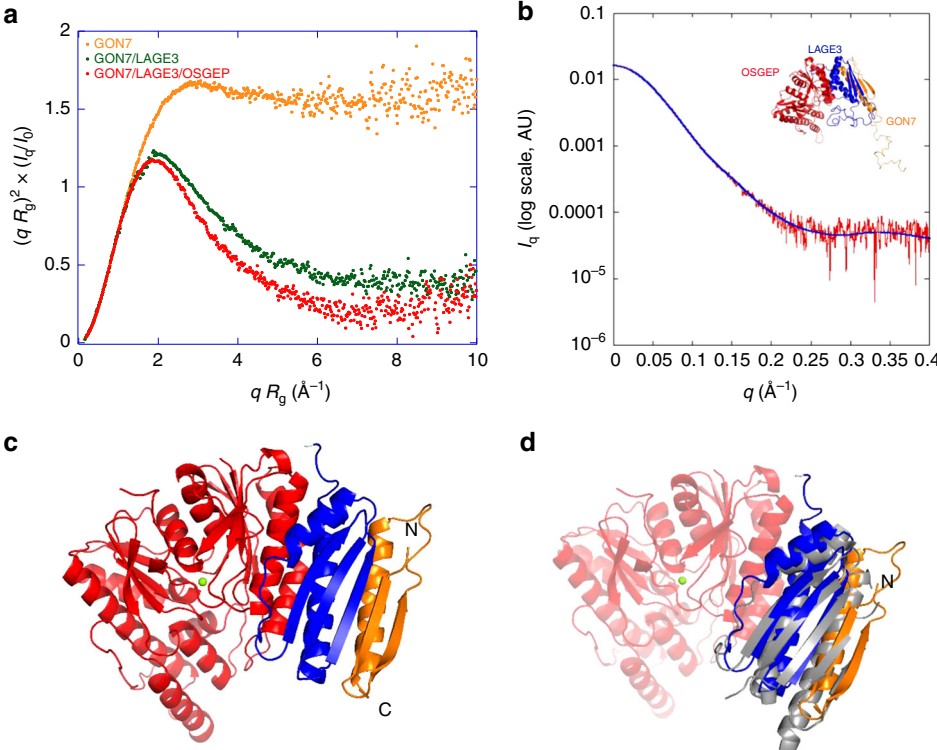

**Fig. 5** Structure of the GON7/LAGE3/OSGEP complex. **a** Normalized Kratky plot of intensity scattering of GON7 (orange) and of the GON7/LAGE3 (green) and GON7/LAGE3/OSGEP complexes (red). $q$: scattering vector, $R_g$ radius of gyration, $I_q$: scattering intensity, $I_0$: scattering intensity at zero angle. **b** Experimental x-ray scattering curve of the GON7/LAGE3/OSGEP complex (red). The blue curve represents the calculated scattering curve for the corresponding crystal structure of the complex. This yielded a good fit with the experimental data ($\chi^2 = 0.33$). The inset shows the BUNCH model. **c** Representation of the crystal structure of the GON7/LAGE3/OSGEP complex: GON7 (gold), LAGE3 (blue), OSGEP (red). The N and C-termini of GON7 are labeled. The crystal lacked density for GON7 beyond residue 50. The active site of OSGEP is highlighted by the $Mg^{2+}$ ion (green). **d** Superimposition of the yeast Gon7/Pcc1 complex (gray) onto GON7/LAGE3/OSGEP. GON7/LAGE3/OSGEP (same color code as in panel **c**)

established that the complex has a 1:1:1 stoichiometry in solution (Fig. 5b, Supplementary Table 4). In contrast with GON7, the dimensionless Kratky plot for GON7/LAGE3/OSGEP shows a bell-shaped curve with a maximum for $qR_g \approx 2$ (Fig. 5a). This shape suggests that the complex is mainly compact, but that disordered regions are still present. In addition, the comparison of the distance distribution functions of GON7/LAGE3/OSGEP and GON7 shows that the latter alone is more extended than the complex (Supplementary Fig. 6b). In full agreement, the $^{15}N$ SOFAST-HMQC NMR spectrum of the complex reveals that about 35 amino-acid residues of GON7 remains flexible and disordered in the complex whereas ~35 crosspeaks vanished upon complex formation. These latter crosspeaks likely correspond to amino-acid residues engaged in the interaction with LAGE3 and thus experiencing extensive line-broadening due to the large molecular size of the complex. We therefore deduced that GON7 is becoming partially ordered upon complex formation with LAGE3/OSGEP and set out to determine its structure by X-ray crystallography. We obtained 1.9 Å resolution diffraction data of the GON7/LAGE3/OSGEP complex (Fig. 5c, d, Supplementary Table 5). The structure could be solved by molecular replacement using our 3D models of OSGEP and LAGE3 (refs. [8,13,23]). LAGE3 contains 60 residues at the N-terminus that are absent in the Pcc1 orthologs from yeast and archaea and that are predicted to be disordered. We did not observe any electron density for this N-terminal peptide, confirming this region indeed lacks a stable structure. We cannot exclude however that partial proteolysis removed this peptide during the long crystallization process. LAGE3 is at the center of the complex, binding on one side to OSGEP and on the other to GON7, which does not directly

interact with OSGEP (Fig. 5c). The structures of the LAGE3 and OSGEP subunits are very similar to their archaeal/fungal Pcc1 and Kae1 orthologs respectively. The two helices of LAGE3 associate with the N-terminal helices of OSGEP into a helical bundle. Only 45% of the GON7 sequence adopts a well-defined structure upon binding to LAGE3, confirming our SAXS- and NMR-based conclusions (Fig. 5a, Supplementary Fig. 6a–d). The N-terminal peptide of GON7 forms a β-hairpin between Met1 and Ser20 and the region between Gly25 and Pro50 forms a helix that lies parallel against the β-hairpin (Supplementary Fig. 7). Electron density for GON7 was absent for residues 21 to 24 and for the region beyond position 50. The C-terminal half of GON7 is highly enriched in acidic and sparse in hydrophobic amino acids and predicted to be unfolded. Despite their very weak sequence similarity (19% identity, 34% similarity), the structures of human GON7 and yeast Gon7 are almost identical (RMSD = 1.41 Å for 45 Cα positions; yeast Gon7 PDB entry: 4WXA) (Fig. 5d, Supplementary Fig. 7). The β-hairpin of GON7 aligns with the β-sheet of LAGE3 to form a continuous five stranded anti-parallel sheet. The helix of GON7 packs in an anti-parallel orientation against the C-terminal helix of LAGE3. The complex is stabilized mainly by the hydrophobic packing of side chains emanating from β1 and α1 of GON7 and α2 and β1 of LAGE3. The association mode between GON7 and LAGE3 is very similar to that of the yeast Pcc1/Gon7 complex, illustrated by their superposition (RMSD = 1.4 Å; Fig. 5d). Structure based sequence alignment between human GON7 and yeast Gon7 shows that only 6 out of 45 ordered residues (12%) are conserved (Supplementary Fig. 7). Compared to human GON7, yeast Gon7 is longer by about 20 residues that were disordered in its structure.

The experimental SAXS curve of the GON7/LAGE3/OSGEP complex in solution is in excellent agreement ($\chi^2 = 0.33$) with the scattering curve calculated on the all-atom model built using the program BUNCH from the crystal structure (see Methods) (Fig. 5b). We therefore conclude that, although sharing very low sequence homology, human GON7 and yeast Gon7 are homologs that interact identically with their respective partners (LAGE3, Pcc1) in the human and yeast KEOPS complex.

**Role of human GON7 in KEOPS complex stability in vivo**. We further explored the deleterious cellular effects of the GAMOS-associated *GON7* mutations. We first confirmed by mass spectrometry analysis that in a human podocyte cell line stably overexpressing either 2HA-tagged GON7 or V5-tagged LAGE3, the four additional KEOPS subunits significantly co-purified with GON7 or LAGE3, respectively, thus confirming that a five-subunit KEOPS complex is present in these renal glomerular cells (Supplementary Fig. 8a). We have previously demonstrated that the majority of GAMOS-associated mutations in genes encoding KEOPS complex components do not affect the intermolecular interactions between the LAGE3/OSGEP/TP53RK/TPRKB subunits[13]. In order to check whether mutations in LAGE3 affected GON7 binding, we co-expressed 2HA-tagged GON7 with WT or mutant V5-tagged LAGE3 in HEK293T cells. Our co-immunoprecipitation experiments demonstrated that the LAGE3 mutations found in GAMOS individuals do not prevent binding to GON7 (Supplementary Fig. 8b). Intriguingly, however, we noticed that co-expression of GON7 with LAGE3 in HEK293T cells led to an increased expression level of GON7, and to a lesser extent of LAGE3, suggesting that the interaction stabilizes both proteins (Fig. 6a). We therefore studied the stability of GON7 and LAGE3 in a time-course experiment using cycloheximide, an inhibitor of protein biosynthesis, in HEK293T cells transiently expressing either 2HA-GON7 or V5-LAGE3 alone or co-expressing both tagged-proteins. When expressed alone, GON7 and LAGE3 protein levels decreased, suggesting both proteins may be unstable in absence of their partner. This is particularly obvious for GON7 whose protein level decreased by half within an hour following cycloheximide addition (Fig. 6b, Supplementary Fig. 9). On the contrary, when co-expressed, we observed an increase of both GON7 and LAGE3 protein levels reflecting an increase in their stability. We wondered whether the absence of GON7 also impacts the stability of the whole KEOPS complex and indeed, we were able to demonstrate that the protein levels of the four KEOPS subunits were decreased in cells of individuals mutated for *GON7*, whereas they were not affected in cells of individuals with *OSGEP* or *WDR73* mutations, the latter being also responsible for a specific subset of GAMOS cases not linked to a t⁶A biosynthesis defect[15] (Fig. 6c, Supplementary Fig. 10a). In addition, we demonstrated that this protein level decrease was not due to transcriptional regulation (Supplementary Fig. 10b). Altogether, these results suggest that the absence of GON7 affects KEOPS stability resulting in a decreased expression level of the four other subunits, which might impact t⁶A levels.

**Discussion**
In this study, we identified mutations in two genes encoding proteins involved in t⁶A biosynthesis in GAMOS patients: *YRDC* encoding the enzyme that synthesizes the TC-AMP intermediate used by the KEOPS complex and *GON7* encoding the fifth subunit of the KEOPS complex. Functional analysis of these specific mutations has revealed that they directly impact t⁶A modification (*YRDC*) and/or affect the stability of the KEOPS complex (*GON7*). These results complement our previous findings and

establish that mutations in all the genes involved in this pathway lead to GAMOS.

All individuals bearing *GON7* or *YRDC* mutations present with the clinical features of GAMOS, similarly to the individuals previously reported to have mutations in the genes encoding the four other KEOPS subunits. In addition, we have expanded the GAMOS phenotype spectrum by describing congenital hypothyroidism to be associated with *YRDC* mutations. Although the two *GON7* mutations encode truncated non-functional proteins, we noticed that they unexpectedly result in a less severe clinical outcome compared to that of individuals affected by mutations in other KEOPS subunit genes or in *YRDC*, for which biallelic null mutations were not found. This suggests that the absence of GON7 has less severe consequences for cell life compared to the other components of the t⁶A biosynthesis pathway, where four out of six are encoded by genes considered to be essential[28]. This less severe clinical outcome correlates with our data showing that GON7 loss of function and depletion in fibroblasts and podocytes, respectively, have globally a weaker effect on t⁶A levels, proliferation, apoptosis, and protein synthesis compared to that of other KEOPS subunits and YRDC mutations or depletion. However, although our data have confirmed that GON7 is a functional homolog of Gon7, the effect of their absence in human and yeast, respectively, is markedly different since in the absence of Gon7, the yeast KEOPS complex has only very low t⁶A activity and cell growth is dramatically affected[3]. Altogether, our data in humans suggest that GON7 is not as essential in humans as in yeast for t⁶A biosynthesis.

Our biochemical and structural data provide a molecular framework to understand the pathophysiological effects of the GAMOS-associated mutations. The structure of the GON7/LAGE3/OSGEP complex shows that GON7 is bound exclusively to the non-catalytic LAGE3 subunit, distant from the catalytic center of OSGEP. It has been shown in vitro that the intact human KEOPS complex has a 1:1:1:1:1 stoichiometry, in contrast with the complex lacking GON7 which has a 2:2:2:2 stoichiometry[7]. The latter stoichiometry has also been observed for the archaeal KEOPS complex, for which no fifth subunit similar to Gon7 has yet been discovered. The Pcc1 subunit constitutes the dimerization unit of archaeal KEOPS[29] and this is also very likely the case for the LAGE3 subunit of human KEOPS in absence of GON7 (ref. [7]). In line with these results, our structure of the GON7/LAGE3/OSGEP complex shows that GON7 competes with LAGE3 for dimerization, explaining the different stoichiometries of the KEOPS complex observed in the absence or presence of GON7. Indeed, GON7 covers a large hydrophobic surface of LAGE3 (Supplementary Fig. 11), which is very likely occupied by another LAGE3 subunit in the context of a homomeric dimer, as observed in the structure of Pcc1 dimer[29]. The exposure of this hydrophobic surface due to the absence of GON7 in the GAMOS patients may affect the solubility and activity of the KEOPS complex, and indeed, our data from experiments on cell lines further indicate that GON7 contributes to the stability of the KEOPS complex and/or to the maintenance of the correct (catalytically active) quaternary structure as evidenced by the decrease in KEOPS subunits protein levels observed in GON7 patient cells. Taken together, our data demonstrated that GON7 impacts the stability of the KEOPS complex therefore having an effect on its enzymatic activity. This is in line with the in vitro data of Sicheri's group showing that in presence of GON7, KEOPS t⁶A activity is potentiated[29] and with our in vivo data showing that t⁶A levels in *GON7*-mutated patient fibroblasts are slightly decreased compared to *YRDC*- and *OSGEP*-mutated fibroblasts.

Although human GON7 and yeast Gon7 have low sequence identity, their structures and interactions with LAGE3 and Pcc1,

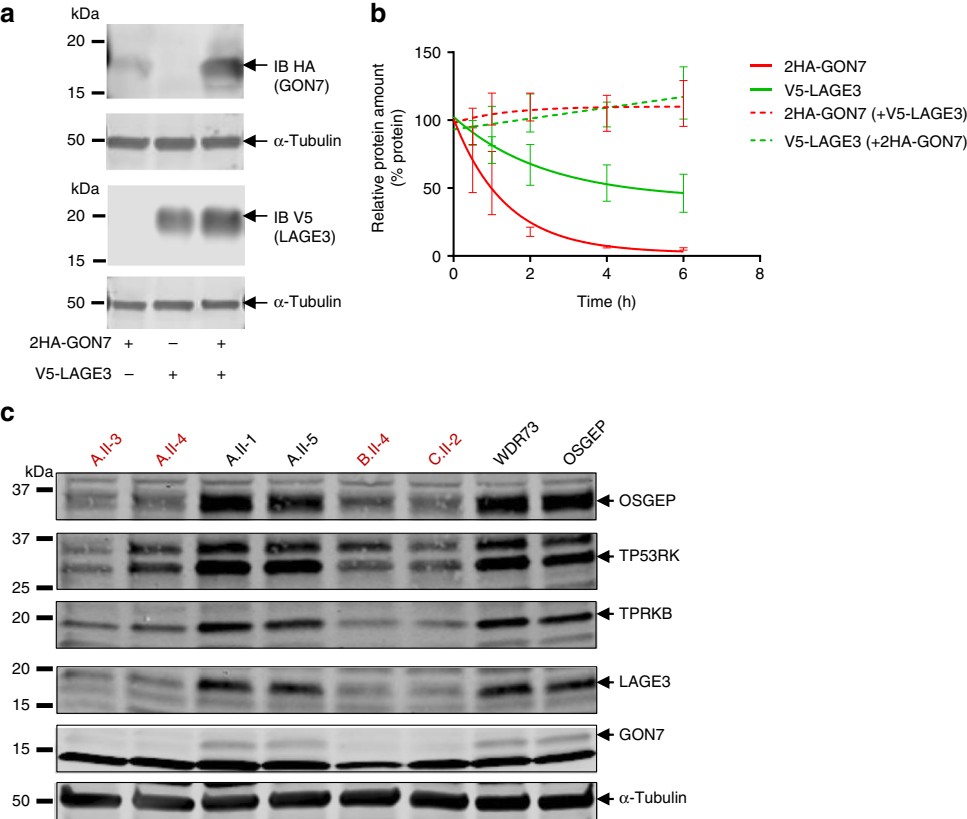

**Fig. 6** Role of GON7 on KEOPS complex stability. **a** Immunoblot analysis of HEK293T cell lysates expressing either 2HA-tagged GON7 or V5-tagged LAGE3 alone or co-expressing both proteins. Anti-HA and anti-V5 antibodies were used to assess GON7 and LAGE3 expression, respectively, with α-tubulin used as loading control. **b** Representation of cycloheximide chase experiments by fitting a one-phase exponential decay curve to experimental data (one representative experiment is shown in Supplementary Fig. 9) (mean ± s.e.m. of $n = 3$ experiments). HEK293T cells were transfected with either 2HA-tagged GON7 or V5-tagged LAGE3 alone or with both proteins before being subjected to treatment with 100 μg/ml cycloheximide for the indicated time points in order to assess rates of protein degradation followed by western blotting of the cell lysates for both proteins with anti-HA and anti-V5 antibodies, respectively. GON7 and LAGE3 protein levels were normalized to those of α-tubulin at each time point. **c** Western blot analysis of protein expression level of the five KEOPS subunits in lymphoblastoid cell lines from two unaffected relatives (A.II-1 and A.II-5), four individuals with the *GON7* mutation p.Tyr*7, one individual with the *OSGEP* mutations p.Arg325Gln and p.Arg280His (individual «N2705» described in Braun et al.[13]), and one individual with GAMOS linked to *WDR73* mutations (individual A.II-4 described in Colin et al.[15]). One representative western blot is shown (three independent experiments were performed). α-Tubulin was used as a loading control. Source data are provided as a Source Data file

respectively, are nearly identical. It is therefore surprising that GON7 could not complement the yeast Δ*gon7* deletion mutant[7]. Comparison of the GON7/LAGE3 and Gon7/Pcc1 complexes shows that the hydrophobic character of the residues at the interface is very well conserved (Supplementary Fig. 7). However quite a few amino-acid substitutions between GON7 and Gon7 might create steric clashes that weaken or disrupt the interaction with Pcc1, explaining the lack of complementation. Nevertheless, the exquisite superposition of GON7 and Gon7 qualifies them as orthologs and confirms that GON7 is the functional fifth subunit of the human KEOPS complex. Such discrepancies between the protein sequence and structure conservation between distant species might be relevant in other protein complex, with their characterization helping to identify new candidate genes for human monogenic disorders.

An increasing number of mutations are being identified in genes encoding tRNA-modifying enzymes that are linked to human neurodevelopmental disorders. Very recently, mutations in *WDR4*, initially described to cause a distinct form of microcephalic primordial dwarfism and brain malformations[30,31], have been identified in individuals with GAMOS[14]. WDR4 is a component of the METTL1/WDR4 holoenzyme, an $N^7$-methylguanosine (m7G) methyltransferase that is responsible for the highly

conserved m7G modification on a specific subset of tRNAs[32,33]. Interestingly, it has been shown that the absence of m7G tRNA modification leads to impaired cell proliferation, neural differentiation, and a decrease in global translation with a less efficient translation of mRNAs involved in cell division and brain development, consistent with the microcephaly and brain anomalies found in individuals with *WDR4* mutations[34]. Similarly, as a consequence of the decrease in t6A levels observed in affected individual's cells, perturbed protein translation could impact the translation of specific mRNA involved in kidney and brain development and/or podocyte/neuron maintenance. It is likely that the requirement for t6A-modified tRNAs levels is dependent of the cell-type and/or cell cycle as has been previously shown in *D. melanogaster* where highly proliferative cells of the wing imaginal discs are more affected by the absence of t6A modification than fully differentiated photoreceptors[35]. Neuronal progenitors that have high mitotic activity probably have higher demands for protein translation, making them more vulnerable to any perturbation in the tight regulation of tRNA modifications. Furthermore, another potential regulatory step to spatiotemporally modulate these tRNA modifications and thus protein translation is the tissue- and developmental stage-specific expression of the tRNA-modifying enzymes[36,37]. YRDC and

KEOPS subunits could be differentially expressed between specific cell-types in the brain (neural progenitors) and the kidney (podocytes) and/or during development/differentiation explaining the tissue involvement and the course of the disease as well as its clinical outcome. Further studies on neuronal/renal progenitors and neurons/podocytes differentiated from induced pluripotent stem cells obtained from individuals with mutations in YRDC and KEOPS subunits will probably provide further insights into the pathogenesis of GAMOS.

Together, our data strongly emphasize the importance and relevance of the t[6]A biosynthesis pathway in the pathogenesis of GAMOS. Further investigations are needed to fully characterize all the KEOPS mutations at the biochemical, structural, and enzyme activity levels to better understand their impact on KEOPS complex-dependent t[6]A biosynthesis activity and how they influence the clinical phenotypes. Genes encoding components of the t[6]A biosynthesis pathway have to be added to the growing list of translation-associated proteins whose loss of function are responsible for rare genetic disorders.

## Methods

**Patients and families**. Written informed consent was obtained from participants or their legal guardians, and the study was approved by the Comité de Protection des Personnes "Ile-De-France II." Genomic DNA samples were isolated from peripheral blood leukocytes using standard procedures.

**Whole-exome sequencing and mutation calling**. We performed whole-exome sequencing using Agilent SureSelect All Exon 51 Mb V5 capture-kit on a HiSeq2500 (Illumina) sequencer (paired-end reads: 2 × 100 bases). Sequences were aligned to the human Genome Reference Consortium Human Build 37 (GRCh37) genome assembly with the Lifescope suite from Life Technologies. Variant calling was made using the Genome Analysis Toolkit pipeline. Then, variants were annotated using a pipeline designed by the Paris Descartes University Bioinformatics platform. We assumed the causal variant: (i) segregates with the disease status, (ii) is novel or has a minor allele frequency <1/1000 in gnomAD, (iii) was not found in >10/2352 projects of our in-house database. Missense variant pathogenicity was evaluated using in silico prediction tools (PolyPhen2, SIFT and Mutation Taster). Sanger sequencing was used to validate the variant identified by exome sequencing and to perform segregation analysis in the families. Sequence were analyzed with the Sequencher software (Gene Codes, Ann Arbor, MI) and positions of mutations were numbered from the A of the ATG-translation initiation codon. For Family G, WES, and SNP-array were performed according to standard diagnostic procedures and WES quality criteria at the UMC Utrecht, the Netherlands. The patient–parent quartet WES with sibling-sharing analysis focused on the regions of homozygosity determined by SNP-array (parents are consanguineous in the eighth degree).

**Plasmids, cell culture, establishment of cell lines**. The following expression vectors were used in this publication: LentiORF pLEX-MCS (Open Biosystems), pESC-TRP with a c-myc tag (Agilent), and pLKO.1-TRC Cloning vector (# Plasmid 10878, Addgene). The LentiORF pLEX-MCS plasmid was modified by site-directed mutagenesis (QuickChange kit, Agilent) to insert one NheI restriction site, and either two copies of the Human influenza hemagglutinin (HA) tag or one copy of the V5 epitope tag allowing epitope-tagging at the N-terminal of the encoded protein. Human full-length GON7, LAGE3, and YRDC cDNA (NM_032490.5, NM_006014.4, and NM_024640.4, respectively) were amplified by PCR from IMAGE cDNA clones (IMAGE 4796574, IMAGE 5485603, IMAGE 5211591, and IMAGE 6147134, respectively), and subcloned into the modified pLEX-MCS plasmid using either SpeI and XhoI (for GON7 and LAGE3) or NheI and XhoI (for YRDC). Human YRDC cDNA was also subcloned into the BamHI and SalI sites of pESC-TRP. Site-directed mutagenesis (QuickChange kit, Agilent) was used to generate the mutations used in this study. An adapted cloning protocol was used to obtain the C-terminal extension found for the YRDC p.Val241Ilefs*72 mutant. For gene silencing, the shRNA sequences described in Supplementary Table 6 were cloned into the lentiviral pLKO.1-TRC Cloning Vector using the AgeI and EcoRI restriction sites. This vector contains a cassette conferring puromycin resistance. All constructs were verified by Sanger sequencing.

The human immortalized podocyte cell line (AB8/13) provided by M. Saleem (University of Bristol, UK) was grown at 33 °C with 7% CO$_2$ in RPMI 1640 medium supplemented with 10% fetal bovine serum, insulin-transferrin-selenium, glutamine, and penicillin and streptomycin (all from Life Technologies), and human primary fibroblasts, obtained from patient skin biopsies, were cultured in OPTIMEM medium supplemented with 10% fetal bovine serum, sodium pyruvate, glutamine, fungizone, and penicillin and streptomycin (all from Life Technologies) at 37 °C with 7% CO$_2$. Obtention and culture of lymphoblastoid cell lines are detailed in Supplementary Methods. Human podocytes stably overexpressing 2HA-GON7 or V5-LAGE3, or transiently depleted for GON7, LAGE3, OSGEP, or YRDC were obtained by transduction with lentiviral particles and subsequent puromycin selection (2 µg/ml). HEK293T cells (ATCC CRL-3216) were transiently transfected using Lipofectamine® 2000 (ThermoFisher Scientific).

**Antibodies and chemical compounds**. The following antibodies were used in the study: mouse anti-α-tubulin (T5168, used at 1:1000), mouse anti-actin (A5316, used at 1:1000), mouse anti-HA (12CA5, at 1/1000), rabbit anti-GON7 (HPA 051832, used at 1:500), rabbit anti-LAGE3 (HPA 036122, used at 1/500), rabbit anti-TPRKB (HPA035712, used at 1:500), rabbit anti-OSGEP (HPA 039751, used at 1/1000), and mouse anti-GAPDH (MAB374, used at 1/2000) from Sigma-Aldrich; mouse anti-V5 (MCA1360, used at 1/1000) from Bio-Rad; rabbit anti-YRDC (PA5-56366, used at 1:500) from ThermoFisher Scientific; rabbit anti-LAGE3 (NBP2-32715, used at 1:1000) and mouse anti-OSGEP (NBP2-00823, used at 1:500) from Novus Biologicals; rabbit anti-TP53RK (AP17010b, used at 1:500) from Abgent. Secondary antibodies for immunoblotting were sheep: anti-mouse and donkey anti-rabbit HRP-conjugated antibodies (GE Healthcare, UK), and IRDye 800CW Donkey anti-rabbit (926-32213) and IRDye 680RD Donkey anti-mouse (926-68072) antibodies (LI-COR). Cycloheximide (C7698), Nuclease P1 (N8630), phosphodiesterase I from snake venom (P3243), and alkaline phosphatase (P4252) were purchased from Sigma-Aldrich.

**Yeast culture and heterologous complementation assay**. Yeast cells were grown at 28 °C in standard rich medium YEPD (1% yeast extract, 2% peptone, 2% glucose) or minimal supplemented media (0.67% YNB, 2% carbon source). Cells were transformed using the lithium acetate method[38]. Media were supplemented with 2% agar for solid media. The S. cerevisiae W303 derived strain, Δsua5::KanMX (YCplac33-SUA5)[39], was used as the host for the complementation assay. For each pESC-TRP plasmid derivative to be tested, three independent clones were selected after transformation and grown on GLU-TRP media. Clones were then streaked onto GAL-TRP containing 0.1% 5-fluoroorotic acid (5-FOA) to counter-select the YCplac33-SUA5 plasmid (containing URA3). After two rounds of selection, clones were checked for their acquired Ura- phenotype, their plasmid content was confirmed by sequencing after plasmid rescue before being finally evaluated for fitness by a 10-fold serial dilution spotted onto GAL-TRP minimal supplemented media. Empty pESC-TRP, pESC-TRP-SUA5, and pESC-TRP-SUA5-myc were used as negative and positive controls, respectively.

**Quantitative real-time PCR**. Total mRNA from knocked down podocytes, primary skin fibroblasts, and LCLs was isolated using Qiagen Extraction RNeasy® Kit and treated with DNase I. One microgram total RNA was reverse-transcribed using Superscript II, according to the manufacturer's protocol (Life Technologies). The relative expression levels of the mRNA of interest were determined by real-time PCR using Power SYBR Green ROX Mix (ThermoFisher Scientific) with specific primers listed in Supplementary Table 7. Samples were run in triplicate and gene of interest expression was normalized to human hypoxanthine-guanine phosphoribosyl transferase (Hgprt). Data were analyzed using the $2^{-\Delta\Delta Ct}$ method.

**Quantification of t[6]A modification**. Yeast tRNAs were extracted and purified from actively growing cells (at $OD_{600nm}$ of approximately $3 \times 10^7$ cells/ml) with phenol induced cell permeabilization, LiCl-selective precipitation, and subsequent ion exchange-chromatography purification on an AXR-80 column (Nucleobond, Macherey-Nagel), according to the manufacturer's instructions. For human primary fibroblasts, the two-step protocol that was applied is detailed in Supplementary Methods. Ten micrograms of yeast tRNAs and approximately 1 µg of human fibroblast tRNAs were then enzymatically hydrolyzed into ribonucleosides with nuclease P1, phosphodiesterase, and alkaline phosphatase, deproteonized by filtration, and finally dried under vacuum according to the protocol of Thuring et al.[40]. t[6]A ribonucleoside was analyzed using the quantitative LC/MS-MS protocol of Thüring et al.[40]. Quantification of t[6]A was performed by integration of the peaks of interest and expressed relative to the total area of the peaks corresponding to the four canonical unmodified ribonucleosides assessed in the same sample for normalization. tRNA extracted from three independent samples were each measured twice (two technical replicates). Detailed information are provided in Supplementary Methods.

**Protein extraction and immunoblotting**. Proteins from KD podocytes, primary fibroblasts, and LCLs were extracted in lysis buffer containing 150 mM NaCl, 50 mM Tris-HCl pH 7, 0.5% Triton-X100 with Complete™ protease inhibitors (Roche), as in Serrano-Perez et al.[41]. Fifty micrograms of proteins were loaded onto acrylamide gels and blotted onto nitrocellulose membranes (Amersham). The membranes were blocked in 1× Tris-buffered saline, 0.1% Tween 20 (TBST) with 5% bovine serum albumin or in Odyssey (LI-COR Bioscience) blocking buffer. Membranes were then incubated with the indicated primary antibodies, washed, and then incubated with either HRP-conjugated or LI-COR IRDye secondary antibodies. Signals were detected using ECL reagents (Amersham Biosciences) and acquired in a Fusion Fx7 darkroom (Vilber Lourmat) or acquired with Odyssey CLx near-infrared fluorescent imaging system (LI-COR Bioscience). Densitometry

quantification was performed either using Bio-1D software or using *Image studio lite* software (version 5.2). Uncropped and unprocessed blots are provided in the Source Data file.

**Immunoprecipitation and cycloheximide chase experiments**. For immunoprecipitation, HEK293T cells were transiently transfected with the adequate plasmids (2HA-tagged GON7, V5-tagged-LAGE3 wild-type (WT) and/or mutants) using calcium phosphate. Forty-eight hours post transfection, cells were lysed in 150 mM NaCl, 25 mM Tris-HCl pH 8, 0.5% Triton with protease inhibitors and HA-tagged GON7 was immunoprecipitated using the μMACS™ Epitope Tag Protein Isolation Kit (Miltenyi Biotec). Briefly, fresh lysates (1–1.5 mg of protein) were incubated either with mouse anti-V5 antibodies, followed by a 30-min incubation with magnetic beads coupled to protein A, or directly with magnetic beads coupled to an HA antibody. Immunoprecipitated proteins were isolated using μMACS® Separation Columns in a magnetic μMACS separator and subsequently eluted with 1× Laemmli buffer. Lysates and immunoprecipitated samples were subjected to immunoblot[41]. To assess rates of protein degradation, HEK293T cells transiently expressing either 2HA-GON7 or V5-LAGE3 alone, or co-expressing both proteins were incubated with cycloheximide at a final concentration of 100 μg/ml for the indicated time periods (0.5, 1, 2, 4, and 6 h). Total protein extracts and immunoblotting were performed as described above. Anti-HA and anti-V5 antibodies were used to reveal GON7 and LAGE3, respectively. Relative GON7 and LAGE3 protein amounts were normalized to those of α-tubulin at each time point.

**Cell proliferation, apoptosis, and protein synthesis assays**. Cell proliferation, apoptosis level, and rates of protein synthesis were assessed in KD podocytes using the CellTiter 96 Aqueous Non-Radioactive Cell Proliferation Assay (MTT) (Promega), the Caspase-3/7 Green detection Reagent (C10423; ThermoFisher Scientific), and the Click-iT HPG Alexa Fluor 488 Protein Synthesis Assays (C10428; ThermoFisher Scientific), respectively, according to the manufacturer's instructions. Detailed information are provided in Supplementary Methods.

**Proteomic studies**. Human podocyte cell lines stably expressing either 2HA-GON7 or V5-LAGE3 were used to perform proteomic studies. 2HA-GON7 and V5-LAGE3 were immunoprecipitated as described in the section above. Eluates were processed according to Braun et al.[13]. Two groups (control IP versus IP HA or IP V5), each containing three biological replicates, were used for statistical analysis. Only proteins that were identified at least three times out of six were retained. A t-test was performed, and the data were represented in a volcano plot (FDR < 0.01, S0 = 2, 250 randomizations).

**Telomeric restriction fragment**. Measurement of the length of the terminal restriction fragments was performed by Southern blotting according to Touzot et al.[42].

**Expression and purification of KEOPS subunits**. All structural work was done using the full-length proteins of GON7, LAGE3, and OSGEP.

For NMR experiments, two vectors were ordered from Genscript (Piscataway, USA) for the expression of either unlabeled his-tagged LAGE3 (vector "pET21a-LAGE3_hisTEV_op") or $^{15}$N-labeled his-tagged GON7 (vector "pET24d-C14_hisTEV_op") whose sequences are shown in Supplementary Table 8. Expression and purification of LAGE3 and $^{15}$N-GON7 and subcomplex LAGE3/$^{15}$N-GON7 preparation are described in detail in Supplementary Methods.

For SAXS or crystallogenesis experiments, preparation of unlabeled GON7 and GON7/LAGE3/OSGEP subcomplex, and co-expression and purification of the KEOPS subunits are detailed in Supplementary Methods and Supplementary Figs. 12 and 13. Fractions of the heterotrimeric GON7/LAGE3/OSGEP complex eluted from size exclusion chromatography (Supplementary Table 9) were then re-loaded onto NiIDA and washed with lysis buffer A supplemented with increasing concentrations of NaCl (0.2; 0.5; 1, and 2 M) in order to remove traces of contaminants. Bound proteins were eluted using three fractions of 2 ml of buffer A supplemented with 100, 200, and 400 mM imidazole and the three subunits complex was concentrated to 8.3 mg/ml for crystallization trials. A unique crystal was obtained using the sitting-drop vapor diffusion method after more than 6 months incubation at 4 °C. The successful condition was composed of 100 nl of protein solution and 100 nl of 30% PEG 4000, 0.1 M Tris-HCl pH 8.5 and 0.2 M magnesium chloride. The crystal was cryo-protected by quick-soaking in reservoir solution supplemented with 30% glycerol prior to flash freezing in liquid nitrogen.

**Modeling and crystal structure determination**. Modeling of YRDC: the Phyre2 and I-tasser webservers both proposed high confidence models for YRDC based on the StSua5 crystal structure despite weak sequence identity between the two species. A 3D model of YRDC was built using the MODELLER software[43]. X-ray diffraction data collection was carried out on beamline Proxima1 at the SOLEIL Synchrotron (Saint-Aubin, France) at 100 K. Data were processed, integrated, and scaled with the XDS program package[44]. The crystal belonged to space group P4₃. The OSGEP and LAGE3 subunits were positioned by molecular replacement with the programs PHASER[45] and MOLREP, implemented in the CCP4 suite[46] using the structures of MjKae1 (PDB ID: 2VWB) and ScPcc1 (PDB ID: 4WX8) as search

models. Residual electron density showed clearly the presence of the GON7 subunit, which was constructed using the program BUCCANEER[46]. The initial structure was refined using the BUSTER program[47] and completed by interactive and manual model building using COOT[48]. The correctness of the assigned GON7 sequence was verified by omit mFo-DFc, 2mFo-DFc, Prime-and-switch electron density maps[49] (Supplementary Fig. 14). One copy of the GON7/LAGE3/OSGEP heterotrimer was present in the asymmetric unit. Data collection and refinement statistics are gathered in Supplementary Table 5. The coordinates have been deposited at the Protein Data Bank (code 6GWJ).

**Small-angle X-ray analysis**. SAXS experiments were carried out at the SOLEIL synchrotron SWING beamline (Saint-Aubin, France). The sample to detector (Aviex CCD) distance was set to 1500 mm, allowing reliable data collection over the momentum transfer range 0.008 Å$^{-1}$ < q < 0.5 Å$^{-1}$ with $q = 4\pi\sin\theta/\lambda$, where $2\theta$ is the scattering angle and $\lambda$ is the wavelength of the X-rays ($\lambda = 1.0$ Å). To isolate the various species in solution, SAXS data were collected on samples eluting from an online size exclusion high-performance liquid chromatography (SEHPLCBio-SEC3Agilent) column and directly connected to the SAXS measuring cell. sixty-five microliters of GON7/LAGE3/OSGEP and GON7 samples concentrated at 1.5 and 6.7 mg/l, respectively, were injected into the column pre-equilibrated with a buffer composed of 20 mM MES pH 6.5, 200 mM NaCl, and 5 mM 2-mercaptoethanol. Flow rate was 300 μl/min, frame duration was 1.0 s, and the dead time between frames was 0.5 s. The protein concentration was estimated by UV absorption measurement at 280 and 295 nm using a spectrometer located immediately upstream of the SAXS measuring cell. A large number of frames were collected before the void volume and averaged to account for buffer scattering. SAXS data were normalized to the intensity of the incident beam and background (i.e. the elution buffer) subtracted using the program FoxTrot[50], the Swing in-house software. The scattered intensities were displayed on an absolute scale using the scattering by water. Identical frames under the main elution peak were selected and averaged for further analysis. Radii of gyration, maximum particle dimensions, and molecular masses were determined using PrimusQT[51] (Supplementary Table 4). The BUNCH program[52] was then used to build atomic models of GON7/LAGE3/OSGEP starting from the crystal structure and by determining the optimal position of the missing regions as to fit the data. In a final step, we substituted the dummy residues of these flexible parts with all-atom descriptions using the programs PD2 and SCWRL4 (ref. [53]). An ultimate adjustment was performed using the program CRYSOL[54]. The modeling was repeated 10 times and the best model was deposited in SASBDB[55] with codes SASDFK8, SASDFM8, and SASDFL8 for GON7, GON7/LAGE3, and GON7/LAGE3/OSGEP, respectively.

**Statistical analyses**. GraphPad Prism 8.0 software was used for the graphical representation and statistical analysis of cell-based data. Results are presented as mean ± s.e.m. of at least $n = 3$ independent experiments. For statistical analysis, data sets comparing more than three conditions (to a control group) were analyzed with ANOVA followed by Dunnett's multiple comparisons test or by using Kruskal–Wallis test followed by a Dunn's multiple comparisons test. Data sets with only two conditions to compare were analyzed using an unpaired t-test or a Mann–Whitney test. $P < 0.05$ was considered statistically significant. A standard confidence interval of 95% was applied in all analyses. Displayed in the figures are the mean values of all technical replicates for each of the independent experiments (displayed as single data points). Black lines indicate the mean values of all independent experiments.

**Reporting summary**. Further information on research design is available in the Nature Research Reporting Summary linked to this article.

## Data availability
The source data underlying Figs. 3c–d, 4 and 6 and Supplementary Figs. 1, 5b, 8b, 9 and 10 are provided as a Source Data file. Patients' consent was not obtained for public deposition of whole-exome sequencing data, but these are available from the corresponding author upon reasonable request. Other data generated during the current study are available from the corresponding authors upon reasonable request. Accession codes for deposited data: crystal structure of GON7/LAGE3/OSGEP (PDB ID: 6GWJ, [https://www.rcsb.org/structure/6GWJ]); SAXS model codes SASDFK8, SASDFM8, and SASDFL8 for GON7, GON7/LAGE3, and GON7/LAGE3/OSGEP, respectively.

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

## Acknowledgements

We are grateful to the patients and their families for their participation. We thank Dr A. Lahoche, Dr. A. Bennour, Dr. C. Loirat, Dr. K. van Gassen, Dr. M. Lilien, and Dr. M. van der Heide-Jalving who helped in the diagnostic process and the follow-up of our patients. We thank G. Froment, D. Nègre, and C. Costa from the lentivector production facility/SFR BioSciences Gerland-Lyon Sud (UMS3444/US8). We acknowledge the use of the biosources of the Necker Imagine DNA biobank (BB-033-00065) and I. Rouvet from CBC BioTec at the CRB-HCL (Hospices Civils de Lyon). We thank Olivier Namy (I2BC, Gif-sur-Yvette, France) for his helpful advices and expertise during the revision process of this article. This work was supported by the Fondation pour le Recherche Médicale (project DEQ2015031682) (to C. Antignac), the European Union's Seventh Framework Programme (FP7/2012, grant 305608 EURenOmics) (to C. Antignac), the Investments for the Future Program (grant ANR-10-IAHY-01) (to C. Antignac), ANR KeoGamo (ANR-18-CE11-0008-01) (to G. Mollet and H.v.T.). This work was supported by the French Infrastructure for Integrated Structural Biology (FRISBI) (ANR-10-INSB-05–01) (to H.v.T.) and the Dutch Kidney Foundation (grant 15OP14) (to A.M.v.E.). S.M. is supported by a Ph.D. grant of the Fondation pour le Recherche Médicale (FRM). P.R. is a scientist from Centre National de la Recherche Scientifique (CNRS).

## Author contributions

R.S., O.G., O.B., E.M., F.N., D.A.B., M.P., C.M., A.M.v.E., F.H., D.M., and C. Antignac performed exome sequencing, bioinformatics analysis of exome sequencing data, whole-exome evaluation, Sanger sequencing, and mutation analysis. R.S., O.B., M.C., S.D, R.N., M.-A.M., B.R., J.B., A.L., D.A.B., A.M.v.E., D.M., F.H., and C. Antignac recruited patients and collected detailed clinical information for the study. N.B. provided MRI from control individuals and critically interpreted MRI images from patients. S.C.-F., R.S., and A.M.v.E. provided and analyzed images of renal histology and electron microscopy. C. Arrondel, J.P., B.C., D.L., G. Martin, E.M., and F.N. performed design of expression vectors used in this study. C. Arrondel generated knockdown cell lines, performed in vitro studies (proliferation, apoptosis, and protein synthesis) in immortalized human podocytes, and performed purification of human tRNAs. C. Arrondel, J.P., G. Menara, L.B., G. Martin, E.M., and F.N. performed cell experiments (co-immunoprecipitation, cycloheximide chase, cell culture), qPCR, and western blot experiments. S.M. performed proteins expression and purifications, cristallogenesis trials, diffraction data collection, 3D structure resolution, SAXS data collection, and analysis. B.C. performed OSGEP/ LAGE3/GON7-his expression and purification, and nucleosides preparation from tRNA samples and YRDC WT and mutant enzymatic assay. D.L. performed yeast complementation studies, expression, and purification of yeast tRNAs. D.D. performed SAXS data collection and analysis. E.L. collected and analyzed NMR experiments. A.-C.B. and S.S. performed HPLC MS/MS t$^6$A modification analysis. G. Mollet, G. Martin, and I.C.G. performed proteomic studies in human podocyte cell lines. P.R. performed telomere restriction-fragment assays. C. Arrondel, S.M., J.P., G. Menara, B.C., D.L., D.D., O.G., O.B., L.B., G. Martin, E.M., F.N., E.L., A.-C.B., S.S., I.C.G., P.R., M.P., C.M., A.M.v.E., D.M., C. Antignac, H.v.T., and G.Mollet contributed to the interpretation of the data. G. Mollet, H.v.T. and C.Antignac conceived and coordinated the study, and wrote the manuscript with the input of B.C., D.L., S.M., and C. Arrondel. All authors critically analyzed and edited the manuscript.

## Additional information

**Competing interests:** F.H. is a cofounder and SAB member of Goldfinch-Bio. The remaining authors declare no competing interests.

Christelle Arrondel[1,24], Sophia Missoury[2,24], Rozemarijn Snoek[3,4], Julie Patat[1], Giulia Menara[1], Bruno Collinet[2,5], Dominique Liger[2], Dominique Durand[2], Olivier Gribouval[1], Olivia Boyer[1,6], Laurine Buscara[1], Gaëlle Martin[1], Eduardo Machuca[1], Fabien Nevo[1], Ewen Lescop[7], Daniela A. Braun[8], Anne-Claire Boschat[9], Sylvia Sanquer[10,11], Ida Chiara Guerrera[12], Patrick Revy[13], Mélanie Parisot[14], Cécile Masson[15], Nathalie Boddaert[16], Marina Charbit[6], Stéphane Decramer[17], Robert Novo[18], Marie-Alice Macher[19], Bruno Ranchin[20], Justine Bacchetta[20], Audrey Laurent[20], Sophie Collardeau-Frachon[21], Albertien M. van Eerde[3,4], Friedhelm Hildebrandt[8], Daniella Magen[22], Corinne Antignac[1,23], Herman van Tilbeurgh[2] & Géraldine Mollet[1]

[1]Laboratory of Hereditary Kidney Diseases, INSERM UMR1163, Université de Paris, Imagine Institute, Paris, France. [2]Institute for Integrative Biology of the Cell (I2BC), CEA, CNRS, Université Paris-Sud, Université Paris-Saclay, Gif-sur-Yvette, France. [3]Department of Genetics, University Medical Center Utrecht, Utrecht, The Netherlands. [4]Center for Molecular Medicine, Utrecht University, Utrecht, The Netherlands. [5]Institut de Minéralogie, de Physique des Matériaux et de Cosmochimie, UMR7590 CNRS/Sorbonne-Université, UPMC, Paris, France. [6]Department of Pediatric Nephrology, AP-HP, Necker Hospital, Paris, France. [7]Institut de Chimie des Substances Naturelles, CNRS UPR2301, Université Paris-Sud, Université Paris-Saclay, Gif-sur-Yvette, France. [8]Department of Medicine, Boston Children's Hospital, Harvard Medical School, Boston, MA, USA. [9]Mass Spectrometry Facility, INSERM UMR1163, Imagine Institute, Paris, France. [10]Service de Biochimie métabolomique et protéomique, Hôpital Necker-Enfants Malades, Paris, France. [11]INSERM UMR-S1124, Université de Paris, Paris, France. [12]Proteomics Platform 3P5-Necker, Université de Paris—Structure Fédérative de Recherche Necker, Inserm US24/CNRS, UMS3633 Paris, France. [13]Inserm UMR1163, Laboratory of Genome Dynamics in the Immune System, Labellisé Ligue contre le Cancer, Université de Paris, Imagine Institute, Paris, France. [14]Genomics Core Facility, Structure Fédérative de Recherche Necker, INSERM U1163 and Inserm US24/CNRS UMS3633, Université de Paris, Paris, France. [15]Bioinformatics Platform, INSERM UMR1163, Université de Paris, Imagine Institute, Paris, France. [16]Department of Pediatric Radiology, and Imagine Institute, INSERM UMR 1163 and INSERM U1000, Université de Paris, Hôpital Necker-Enfants Malades, Paris, France. [17]Department of Pediatric Nephrology-Internal Medicine, Purpan Hospital, Toulouse, France. [18]Pediatric Nephrology Unit, University Hospital of Lille, Lille, France. [19]Department of Pediatric Nephrology, AP-HP, Robert Debre Hospital, Paris, France. [20]Service de Néphrologie, Rhumatologie et Dermatologie pédiatriques, Hospices Civils de Lyon, Hôpital Femme-Mère-Enfant, Centre de référence de maladies rénales rares, Université de Lyon, Bron, France. [21]Department of Pathology, Hospices Civils de Lyon-Hôpital Femme-Mère-Enfant, Claude Bernard Lyon 1 University, Bron, France. [22]Pediatric Nephrology Institute-Rambam Health Care Campus-Technion Faculty of Medicine, Haifa, Israel. [23]Department of Genetics, AP-HP, Necker Hospital, Paris, France. [24]These authors contributed equally: Christelle Arrondel, Sophia Missoury.

