## [Peer Review File · Nature Communications]

Reviewers' Comments:

Reviewer #1:

Remarks to the Author:

The manuscript by Arrondel et al. provides an interesting study linking mutations in GON7 and YRDC with defects in t6A tRNA modification leading to Galloway-Mowat syndrome.

It is a well written and elaborated study ascertaining the crucial role of the t6A modification in GAMOS pathogenesis. In addition to numerous biochemical experiments, the authors performed homology modelling of human YRDC and analyzed structurally the GON7/LAGE3/OSGEP sub-complex of KEOPS multi-domain assembly. Interestingly, the intrinsically disordered GON7 protein became partially structured upon interacting with its physiological partner LAGE3.

However, the description of the structural work lacks some important details which in my eyes are necessary to reliably assess its quality prior to publication.

1. The presented model of human YRDC with bound threonylcarbamoyladenylate is based on *Sulfolobus tokodaii* Sua5 complex sharing only 20% of sequence identity. The authors do not describe any details regarding verification of sequence alignment, modelling of insertions and/or deletions as well as docking of threonylcarbamoyladenylate/Mg ion (if performed at all). Which software package has been used for modelling/docking purpose? It should be mentioned and also cited in the manuscript.

In addition, based on the depicted model (Figure 1e) the reader might get a wrong impression that human YRDC is a single domain protein, as in contrast, the Sua5 protein is composed of two domains. If the second domain has been removed for clarity reasons it should be stated.

Also, comparison of 4E1B structure with the presented model of human YRDC surprisingly reveals that the structures are much more similar than one could have predicted based on the 20% sequence identity. This questions the validity of performed homology modelling (models provided by "black-box" servers shall not be blindly trusted and not published without validation).

However, for the purpose of this study, it should be also sufficient to mark positions of the mutated residues directly on Sua5 protein and state it accordingly. This will resolve issues related to homology modelling of human YRDC and generating its complex with threonylcarbamoyladenylate.

2. The presented SAX analysis of GON7/LAGE3/OSGEP sub-complex reveals that some disordered regions of presumably GON7 and/or N-terminal fragment of LAGE3 are present. However, the authors do not state whether full lengths or truncated constructs have been used for the analysis. The Supplemental Table 6 provides the sequences, however, without an additional effort, one can not estimate if full-length components have been used for crystallization experiments.

4. The authors describe that no electron density map has been observed for the 60 aa long N-terminal fragment of LAGE3 and about 50 aa long disordered part of GON7 – assuming full lengths proteins have been used for crystallization. The authors conclude that these disordered fragments must have been disordered. However, the lack of electron density map does not prove that these two unstructured fragments are present in the crystal lattice. Taking into account the resolution of 1.9 Å and reported crystallographic R factors, the most likely scenario is proteolytic cleavage of the unstructured fragments. This could explain the presence of a single crystal which has been

obtained after 6 months. Thus the statement concerning the presumable unstructured parts should be accordingly corrected as an additional likely explanation is possible. Ideally, the authors could prove/disprove the proteolytic cleavage if the crystallization droplet is still available.

5. The structure solution part lacks some details, e.g.: software used for structure solution and refinement seems to be not cited. There is no doubt that three components of KEOPS complex have been crystallized. However, taking into account a very low sequence similarity between human GON7 and yeast Gon7, sequence assignment could have been challenging based on the electron density map originating from Molecular Replacement. How was the modelled sequence of GON7 protein verified? The authors should provide a few snapshots of the atomic model and corresponding omit mFo-DFc/2mFo-DFc/Prime-and-switch electron density map allowing the reader to verify the correctness of the sequence assignment. This is a critical issue as most of GON7 residues reveal a rather poor fit to the electron density map.

6. The authors discussed the similarity of GON7/Gon7 binding to their partners: LAGE3 and Pcc1, respectively. However, the assumption that GON7 could complement the yeast Δ gon7 deletion mutant solely based on a superposition of two structural models is far too simplistic. Therefore the statement: Line 366: "The reasons for this cannot be readily deduced from the structural data since the superposition of the respective complexes suggests that human GON7 should be able to replace the yeast Gon7 subunit within the yeast KEOPS complex" is most likely not correct.

Did the authors try to analyze the interacting residues and their conservation between human and yeast? Having a structural model it should be easily done.

Minor remarks:

Line: 140: "and affect/weaken the structure of the protein should be replaced with "and destabilize the structure of the protein"

Line: 262 Despite their very weak sequence similarity, the structures of human GON7 and yeast Gon7 are almost identical. → What are the sequence identity and similarity between the two proteins?

Line 269 Pcc1/Gon7 complex, illustrated by their perfect superposition (RMSD= 1.4 Å; Fig.5d). → The word "perfect" seems to be too optimistic. Omitting it will simply state the fact.

Supplementary Table 3 lacks units (B-factors, RMSDs bonds, angles). The authors should replace the incorrect abbreviation RMS with RMSD.

Reviewer #2:

Remarks to the Author:

In their manuscript "Defects in t6A tRNA modification due to GON7 and YRDC mutations lead to Galloway-Mowat syndrome" Arrondel et al. report mutations found in KEOPS subunits resulting in Galloway-Mowat syndrome and characterize their effect on a cellular level. They also report the crystal and solution structures of GON7/LAGE3/OSGEP complex. The paper is well written and

easily understandable. This review focuses on the SAXS part of the reported results. The authors used SEC-SAXS to collect data on both GON7 and GON7/LAGE3/OSGEP complex and could nicely show that the IDP adapts a folded conformation in the complex. There are, however, a few minor issues with the way the results are reported that need to be addressed:

- The BUNCH model built to fit the data is not shown at all, nor is any indication given on how different it is to the crystallographic model. The model needs to be shown and should also be deposited in a relevant database, e.g. sasbdb.

- Table S2 does not correspond to the current reposting guidelines endorsed by the IUCr (<https://journals.iucr.org/d/issues/2017/09/00/jc5010/>). For the Guinier-approximation based values (R_g , mass), an error should be reported, taking into account the uncertainty of the concentration determination.

In addition, a log-log or log-lin representation of the GON7 data and the Guinier plot are missing.

- References are missing for the crystallographic and SAXS instruments and software. In addition, the synchrotron, beamlines and beamline scientists should be acknowledged (<https://www.synchrotron-soleil.fr/en/users/after-experiment>)

Reviewer #3:

Remarks to the Author:

Four components of KEOPS complex were previously reported as the core enzyme for tRNA t6A modification, and the deficiency of either of the 4 proteins was responsible for GAMOS (nephropathy and microcephaly). Recently, a 5th component of KEOPS complex, GON7, was identified, and recombinant GON7 was shown to promote in vitro t6A reconstitution. In this manuscript, the authors showed that like the previously well-characterized 4 components of KEOPS complex, GON7 mutation causes GAMOS, which is likely because GON7 promotes stabilization of KEOPS complex. Also, the structure of GON7-LAGE3-OSGEP complex establishes GON7 as a true component of KEOPS complex. However, the manuscript lacks further major insights and conceptual advances, and thus would not attract a lot of readership. To gain important insights, it would be nice if the authors could address questions such as the followings.

1. Major questions:

t6A is a ubiquitous tRNA modification that is observed in all organs. Why is the phenotype of GAMOS seen primarily in kidney and brain? Does the kidney and brain (or their progenitor cell) translation require more decoding by t6A-containing tRNAs? In the public NGS databases, there are ribosome profiling NGS data of different organs (at least mice organs). So, it might be nice to analyze such data to see if brain and kidney translate more codons that are decoded by t6A-containing tRNAs, compared to other organs such as liver. Ideally, the authors should perform ribosome profiling to compare translation in wild-type cell, GON7 KO cell, and YRDC KO cell.

2. Relatively minor yet a question that needs to be addressed:

Patient-derived GON7-deficient culture cells (with Tyr7Stop GON7) did not show decrease of t6A (Fig. 3d). Yet, GAMOS patients show mild but evident GAMOS phenotypes. Can the authors find situations in which GON7-deficient cells show t6A decrease? For example, I wonder if using cell culture media with lower threonine concentration would lead to decreased t6A level exclusively in patient-derived cells and not in wild-type cells, because threonine is the substrate for t6A modification.

3. Minor point:

In page 8, line 177, the authors wrote that YRDC protein from 4 nt deletion mutant (Val24Ile frameshift) "was barely detectable by western blot, suggesting that it is likely being degraded by an intracellular proteolytic machinery (Fig 3b)". However, generally, when there is a frameshift mutation, it causes the appearance of a premature termination codon, which induces nonsense-mediated mRNA decay, rather than protein degradation.

Reviewer #4:

Remarks to the Author:

The article "Defects in t6A tRNA modification due to GON7 and YRDC mutations lead to Galloway-Mowat syndrome" by Arrondel et al. presents a comprehensive study of two proteins involved in the t6A biosynthesis, namely YRDC and GON7, one of the five KEOPS proteins in humans. Mutations in these proteins can lead to the Galloway-Mowat syndrome (GAMOS), a rare neurological disorder. Overall the manuscript is well-written and concise and shows in-depth in vivo and vitro data.

The results in the manuscript are essentially separated in two parts, a genetic/biochemical part and a structural investigation.

In the first part, through whole-exome sequencing of several patients with GAMOS, disease-related mutations in GON7 and YRDC were identified. Mutations in YRDC result in a more severe clinical outcome, suggesting that mutations in GON7 can be better compensated, as supported by cellular assays and the direct quantification of t6A content. In the second part, the authors seek to explain the in vivo/vitro findings for YRDC and GON7 on a structural biology level. They investigate the structural basis via homology models as well as NMR, SAXS and X-ray crystallography.

In the following, I will elaborate particularly on the second part, which requires some major revisions:

- 1) The in silico model of human YRDC must be treated with considerable caution as it relies on a crystal structure from yeast with only 20% sequence identity. Please add details on how the homology model was created.
- 2) The conclusions based on the structural model for YRDC are not supported by experimental data, at best the in silico model might give some vague indications. Since YRDC was identified as the clinically more significant one compared to GON7, some experimental, structural data should be included for YRDC as well. Ideally, of course, solving the atomic resolution structure of human YRDC can allow for a more detailed understanding of the molecular mechanism behind the dramatic impact of various YRDC mutations on a clinical level. Alternatively, the folding/stability can be also probed by NMR spectroscopy, which was already employed in this study as a high-resolution technique. The NMR investigation of different YRDC mutants can be conclusive in terms of folding/stability, for example, when comparing 2D ¹H,¹⁵N hetNOE, HSQCs/HMQCs or even 1D ¹H spectra. Potentially, this would significantly strengthen the in silico model.
- 3) The authors investigate the folding of human GON7 via NMR spectroscopy and, therefore, recorded 2D ¹H,¹⁵N HMQC spectra. I agree, that the GON7 spectrum strongly suggests, that GON7 is unstructured based on the narrow chemical shift dispersion (SI Figure 4a, blue). But I do not understand the spectrum of GON7+LAGE3 (SI Figure 4a, red). The authors cautiously state, that "adding non-labelled LAGE3 to the sample caused many chemical shift displacements, suggesting GON7 interacts with LAGE3". Comparing both 2D spectra reveals that peaks almost exclusively disappear, in fact, there are very few shifts. This is surprising. One would expect, that a molecular interaction between GON7 and LAGE3 necessarily implies (partial) folding of GON7. However, the GON7 spectra with and without LAGE3 look essentially the same, there are no additional peaks appearing in the spectral regions indicative of secondary structure elements. How do the authors explain this? How does the spectra look at low contour levels close to the noise level? Are the peaks from folded regions simply clipped by the high contour threshold? The authors should also add the gel filtration profiles of the purified GON7+LAGE3 sample for the NMR study. I am stressing this as it would be a significant point to make, if the presence of LAGE3 induces in vitro folding of GON7. Or, the alternative, does partial folding require also the OSGEP protein? For the orthologous proteins in yeast, the dimeric complex of Gon7 and Pcc1 was sufficient to induce partial folding and allowed to determine the crystal structure. Is there a major mechanistic difference between human and yeast?

4) Please provide the experimental X-ray scattering curves for GON7 and GON7+LAGE3. Is there a concentration dependence, which indicates aggregation? That could also explain the loss of signals in the NMR spectrum.

5) The authors present a high-resolution crystal structure of the GON7/LAGE3/OSGEP complex, which reveals a direct interaction of GON7 to LAGE3 but not with the catalytic OSGEP protein. Strikingly, the binding mode of human GON7/LAGE3 is structurally conserved with respect to the yeast Gon7/Pcc1 despite the very low sequence identity. With the structure at hand, can the authors bring the clinical impact of the GON7 mutations more into a structural context? How do mutations in GON7 impair the KEOPS complex? The manuscript still lacks a link between disease-related mutations and the complex structure. This should be made more clear as the structure is somewhat unattached.

6) Minor point: In the experimental details given in the Supplementary Information the authors describe in their expression protocol that they have used 0.5 μ M IPTG for induction. I believe this should read 0.5 mM IPTG.

It is obvious that it is a major effort combining heterogeneous results as presented in this manuscript. The authors show some very interesting in vivo and in vitro findings, however, the structural biology part, which I can mainly assess, requires major revisions. It still lacks a more clear connection of the complex structure to the rest of the manuscript and further conclusions on how this structure can be helpful, e.g. in the context of medical treatment.

Reviewer #5:

Remarks to the Author:

In the manuscript "Defects in t6 1 A tRNA modification due to GON7 and YRDC mutations lead to Galloway Mowat syndrome", Arrondel et al describe 7 families with mutations in GON7 and YRDC leading to Galloway Mowat Syndrome, a severe form of syndromic nephrotic syndrome. The work is well presented and seems robust. Prior work of the some of the same authors on genes in the same KEOPS complex provides additional support to these findings.

Only few minor details will need to be addressed.

I would like the authors to add additional population allele frequencies for the likely pathogenic variants, since the families all come from areas of the world that were not well sampled in currently available databases. I am confident that the results presented here are robust, but it would be important to see population genetics data from ethnically matched controls, just to make sure that none of the variants reported here are observed in homozygous or compound heterozygous state in these populations.

Reviewers' comments and our answers:

Reviewer #1 (Remarks to the Author):

The manuscript by Arrondel et al. provides an interesting study linking mutations in GON7 and YRDC with defects in t6A tRNA modification leading to Galloway-Mowat syndrome.

It is a well written and elaborated study ascertaining the crucial role of the t6A modification in GAMOS pathogenesis. In addition to numerous biochemical experiments, the authors performed homology modelling of human YRDC and analyzed structurally the GON7/LAGE3/OSGEP sub-complex of KEOPS multi-domain assembly. Interestingly, the intrinsically disordered GON7 protein became partially structured upon interacting with its physiological partner LAGE3.

However, the description of the structural work lacks some important details which in my eyes are necessary to reliably assess its quality prior to publication.

1. The presented model of human YRDC with bound threonylcarbamoyladenylate is based on *Sulfolobus tokodaii* Sua5 complex sharing only 20% of sequence identity. The authors do not describe any details regarding verification of sequence alignment, modelling of insertions and/or deletions as well as docking of threonylcarbamoyladenylate/Mg ion (if performed at all). Which software package has been used for modelling/docking purpose? It should be mentioned and also cited in the manuscript.

In addition, based on the depicted model (Figure 1e) the reader might get a wrong impression that human YRDC is a single domain protein, as in contrast, the Sua5 protein is composed of two domains. If the second domain has been removed for clarity reasons it should be stated.

Also, comparison of 4E1B structure with the presented model of human YRDC surprisingly reveals that the structures are much more similar than one could have predicted based on the 20% sequence identity. This questions the validity of performed homology modelling (models provided by “black-box” servers shall not be blindly trusted and not published without validation).

However, for the purpose of this study, it should be also sufficient to mark positions of the mutated residues directly on Sua5 protein and state it accordingly. This will resolve issues related to homology modelling of human YRDC and generating its complex with threonylcarbamoyladenylate.

We made a structure based sequence alignment of human YRDC, *E. coli* YrdC, *Sulfolobus tokodaii* and *Pyrococcus abyssi* Sua5 (Supplementary Fig.2), showing that YRDC and the YrdC domains of Sua5 have very few deletions or insertions (see main text lines 142-147).

Human YRDC is a single domain protein in contrast with the *S. tokodaii* Sua5 which has an extra (Sua5) domain. This was indeed not explicitly mentioned in the manuscript but it has now been explained in the introduction of the revised version. (lines 83-85).

We obtained the first 3D models using the web servers Phyre2 and I-tasser. Both webservers proposed very similar high-confidence models of YRDC based on the *S. tokodaii* Sua5 structure, despite the weak sequence similarity. The model that was used for the figure was constructed with the program Modeller (information added in the material section, lines 597-600)

Indeed, the structures of the proteins involved in TC-AMP intermediate in general are very well conserved. To illustrate this point, we superposed the experimental structure of YrdC from *E. coli* (PDB ID 2MX1) with those from Sua5 from *S. tokodaii* (PDB ID 3AJE) (28% sequence identity) and *P. abyssi* (PDB ID 6F87) (28% sequence identity) and obtained RMSD values of 2.8 Å.

The final purpose of the modeling of the YRDC protein was mainly to show the positions of the GAMOS mutations relative to the active site region. No docking was carried out for the threonylcarbamoyladenylate (TC) intermediate, the model of the complex was obtained by superposing the *S. tokodaii* Sua5 TC-AMP complex onto YRDC.

2. The presented SAX analysis of GON7/LAGE3/OSGEP sub-complex reveals that some disordered regions of presumably GON7 and/or N-terminal fragment of LAGE3 are present. However, the authors do not state whether full lengths or truncated constructs have been used for the analysis. The Supplemental Table 6 provides

the sequences, however, without an additional effort, one can not estimate if full-length components have been used for crystallization experiments.

The full length proteins have been used for the crystallography and SAXS experiments. This is now explicitly mentioned in the materials and methods section (line 578).

4. The authors describe that no electron density map has been observed for the 60 aa long N-terminal fragment of LAGE3 and about 50 aa long disordered part of GON7 – assuming full lengths proteins have been used for crystallization. The authors conclude that these disordered fragments must have been disordered. However, the lack of electron density map does not prove that these two unstructured fragments are present in the crystal lattice. Taking into account the resolution of 1.9 Å and reported crystallographic R factors, the most likely scenario is proteolytic cleavage of the unstructured fragments. This could explain the presence of a single crystal which has been obtained after 6 months. Thus the statement concerning the presumable unstructured parts should be accordingly corrected as an additional likely explanation is possible. Ideally, the authors could prove/disprove the proteolytic cleavage if the crystallization droplet is still available.

We agree with that comment. Unfortunately, crystallization droplet was not available. We thus have added in the text that partial proteolysis could have taken place during the crystallization as an alternative explanation for the absence of electron density for the first 60 residues of LAGE3. (lines 283-284)

5. The structure solution part lacks some details, e.g.: software used for structure solution and refinement seems to be not cited. There is no doubt that three components of KEOPS complex have been crystallized. However, taking into account a very low sequence similarity between human GON7 and yeast Gon7, sequence assignment could have been challenging based on the electron density map originating from Molecular Replacement. How was the modelled sequence of GON7 protein verified? The authors should provide a few snapshots of the atomic model and corresponding omit mFo-DFc/2mFo-DFc/Prime-and-switch electron density map allowing the reader to verify the correctness of the sequence assignment. This is a critical issue as most of GON7 residues reveal a rather poor fit to the electron density map.

We have added more information on how the structure was solved in the materials section (lines 603-613). Briefly, the OSGEP and LAGE3 subunits were positioned by molecular replacement with the programs PHASER and MOLREP, implemented in the CCP4 suite, using the structures of Methanococcus jannaschii Kae1 (PDB ID: 2VWB) and Saccharomyces cerevisiae Pcc1 (PDB ID: 4WX8) as search models. After successful placement of OSGEP and LAGE3, the residual electron density unambiguously showed the presence of the GON7 subunit. The program BUCCANEER was used to build GON7 into the residual density map. We have calculated omit mFo-DFc/2mFo-DFc/Prime-and-switch maps, confirming that the sequence assignment was correct (shown as Supplementary Fig.12)

6. The authors discussed the similarity of GON7/Gon7 binding to their partners: LAGE3 and Pcc1, respectively. However, the assumption that GON7 could complement the yeast Δ gon7 deletion mutant solely based on a superposition of two structural models is far too simplistic. Therefore the statement: Line 366: The reasons for this cannot be readily deduced from the structural data since the superposition of the respective complexes suggests that human GON7 should be able to replace the yeast Gon7 subunit within the yeast KEOPS complex” is most likely not correct.

Did the authors try to analyze the interacting residues and their conservation between human and yeast? Having a structural model it should be easily done.

The hydrophobic nature of the Gon7/Pcc1 and GON7/LAGE3 interfaces is well conserved. However, there are sequence variations between the two complexes at the level of these interfaces. GON7 residues in interaction with LAGE3 are now marked in the sequence alignment of Supplementary Fig.7. We noticed a few putative steric clashes when modeling for instance Gon7 onto LAGE3 or vice versa GON7 onto Pcc1 among positions marked with blue stars in Supplementary Fig.7. We changed the text accordingly (lines 395-403).

Minor remarks:

Line: 140: and affect/weaken the structure of the protein should be replaced with “and destabilize the structure of the protein”

This has been corrected (line 149).

Line: 262 Despite their very weak sequence similarity, the structures of human GON7 and yeast Gon7 are almost identical. → What are the sequence identity and similarity between the two proteins?

Using structure based alignment, we obtain 19% identity and 34 % similarity (information added in the text line 295)

Line 269 Pcc1/Gon7 complex, illustrated by their perfect superposition (RMSD= 1.4 Å; Fig.5d). → The word “perfect” seems to be too optimistic. Omitting it will simply state the fact.

“perfect” has been removed (line 302)

Supplementary Table 3 lacks units (B-factors, RMSDs bonds, angels). The authors should replace the incorrect abbreviation RMS with RMSD.

Table was corrected

Reviewer #2 (Remarks to the Author):

In their manuscript “Defects in t6A tRNA modification due to GON7 and YRDC mutations lead to Galloway-Mowat syndrome” Arrondel et al. report mutations found in KEOPS subunits resulting in Galloway-Mowat syndrome and characterize their effect on a cellular level. They also report the crystal and solution structures of GON7/LAGE3/OSGEP complex. The paper is well written and easily understandable. This review focuses on the SAXS part of the reported results. The authors used SEC-SAXS to collect data on both GON7 and GON7/LAGE3/OSGEP complex and could nicely show that the IDP adapts a folded conformation in the complex. There are, however, a few minor issues with the way the results are reported that need to be addressed: 1-The BUNCH model built to fit the data is not shown at all, nor is any indication given on how different it is to the crystallographic model. The model needs to be shown and should also be deposited in a relevant database, e.g. sasbdb.

A figure showing the BUNCH model was added as an inset to Fig.5b. Models have been deposited at the SASBDB (codes: SASDFK8, SASDFM8 and SASDFL8 for GON7, GON7/LAGE3 and GON7/LAGE3/OSGEP, respectively) and mentioned in material section of main text (lines 640-642)

- Table S2 does not correspond to the current reposting guidelines endorsed by the IUCr (<https://journals.iucr.org/d/issues/2017/09/00/jc5010/>). For the Guinier-approximation based values (R_g, mass), an error should be reported, taking into account the uncertainty of the concentration determination. In addition, a log-log or log-lin representation of the GON7 data and the Guinier plot are missing.

Supplementary Table 2 was modified to follow the IUCr guidelines. The Log/log plot and Guinier plots were added to Supplementary Fig.6 c and d, respectively.

- References are missing for the crystallographic and SAXS instruments and software. In addition, the synchrotron, beamlines and beamline scientists should be acknowledged (<https://www.synchrotron-soleil.fr/en/users/after-experiment>)

References for the SAXS data treatment were added to the legend of Supplementary Table 2, references for crystallography were added to the main text (lines 603 to 613). Beamline scientists were acknowledged.

Reviewer #3 (Remarks to the Author):

Four components of KEOPS complex were previously reported as the core enzyme for tRNA t6A modification, and the deficiency of either of the 4 proteins was responsible for GAMOS (nephropathy and microcephaly). Recently, a 5th component of KEOPS complex, GON7, was identified, and recombinant GON7 was shown to promote in vitro t6A reconstitution. In this manuscript, the authors showed that like the previously well-characterized 4 components of KEOPS complex, GON7 mutation causes GAMOS, which is likely because GON7 promotes stabilization of KEOPS complex. Also, the structure of GON7-LAGE3-OSGEP complex establishes GON7 as a true component of KEOPS complex. However, the manuscript lacks further major insights and conceptual advances, and thus would not attract a lot of readership. To gain important insights, it would be nice if the authors could address questions such as the followings.

1. Major questions:

t6A is a ubiquitous tRNA modification that is observed in all organs. Why is the phenotype of GAMOS seen primarily in kidney and brain? Does the kidney and brain (or their progenitor cell) translation require more decoding by t6A-containing tRNAs? In the public NGS databases, there are ribosome profiling NGS data of different organs (at least mice organs). So, it might be nice to analyze such data to see if brain and kidney translate more codons that are decoded by t6A-containing tRNAs, compared to other organs such as liver. Ideally, the authors should perform ribosome profiling to compare translation in wild-type cell, GON7 KO cell, and YRDC KO cell.

We agree with the reviewer that why alterations in ubiquitous proteins lead to organ specific diseases is a major and fascinating question that goes far beyond Galloway-Mowat syndrome.

As proposed by the reviewer, we tried to analyze ribosome profiling data provided in the public databases. However, we were unable to do a meaningful analysis as datasets available for brain, kidney or liver were too different in their experimental design (not the same animal, same genetic background, same developmental stage, same experimental conditions). Nevertheless, one study (Castelo-Szekely et al., 2017, doi: 10.1186/s13059-017-1222-2) allowed the comparison between kidney and liver translation at adult stage (from the same mouse). Ribosome profiling data obtained from this study were analyzed by Olivier Namy, an expert in ribosome profiling (I2BC, Gif-sur-Yvette, France). No enrichment in ANN codon could be detected in the genes encoding proteins expressed in the kidney compared to those expressed in the liver. In parallel, we performed another analysis using the human protein atlas database (<https://www.proteinatlas.org/humanproteome/tissue>). We took the tissue-enriched genes (presenting at least five-fold higher mRNA levels in a particular tissue as compared to all other tissues) in the brain (419 genes), in the kidney (54 genes), in the heart (28 genes) and in the liver (157 genes) and calculated their percentage in ANN codons. Again, we did not find any enrichment in ANN codons in these tissue-enriched genes. These preliminary data are of course not sufficient to draw any conclusion and thus we prefer not to include them in the manuscript.

It is clear that ribosome profiling is unavoidable to better understand the molecular and cellular alterations due to t⁶A deficiency, but it is out of the scope of the present paper. Indeed, the only GON7 and YRDC KO cells currently available are fibroblasts and/or lymphoblastoid cell lines (LCLs) that we do think are not the more relevant cell types to use, and we plan to perform the ribosome profiling experiments (in collaboration with Olivier Namy) in more relevant cellular models such as podocytes, the cells affected in the kidney, and neural progenitor cells (NPCs) that will be differentiated from iPSCs (induced pluripotent stem cells) from patients with GON7 or OSGEP mutations.

2. Relatively minor yet a question that needs to be addressed:

Patient-derived GON7-deficient culture cells (with Tyr7Stop GON7) did not show decrease of t6A (Fig. 3d). Yet, GAMOS patients show mild but evident GAMOS phenotypes. Can the authors find situations in which GON7-deficient cells show t6A decrease? For example, I wonder if using cell culture media with lower threonine concentration would lead to decreased t6A level exclusively in patient-derived cells and not in wild-type cells, because threonine is the substrate for t6A modification.

Although not significant, we observed a slight decrease in t⁶A levels in patient GON7-deficient fibroblasts. As mentioned above, we strongly believe that differences in t⁶A levels will be more obvious in cells and/or tissues relevant to the disease (ie podocytes, NPC), which we could check only when we get the iPSC cells. If no difference could be found in these podocytes/NPC cells, we will use the elegant reviewer suggestion to try to decrease their t⁶A level.

3. Minor point:

In page 8, line 177, the authors wrote that YRDC protein from 4 nt deletion mutant (Val241Ile frameshift) “was barely detectable by western blot, suggesting that it is likely being degraded by an intracellular proteolytic machinery (Fig 3b)”. However, generally, when there is a frameshift mutation, it causes the appearance of a premature termination codon, which induces nonsense-mediated mRNA decay, rather than protein degradation.

Actually, both in the experimental conditions (western blot was performed on protein lysates from yeast overexpressing the human YRDC cDNA) or *in vivo*, the c.721_724 del (p. Val 241 Ile fs*72) mutation is predicted to escape RNA decay (no introns in the cDNA construct in the former case and stop codon in the last exon in the latter) (see Maquat et al., 2004, doi:10.1038/nrm1310).

To confirm these predictions, we performed RT-qPCR on fibroblast cell extracts from patients with YRDC mutations and demonstrated the presence of YRDC transcripts at the same levels as in the controls. These new data have been added to Supplementary Fig.1 and mentioned in the main text (lines 140-142).

Reviewer #4 (Remarks to the Author):

The article "Defects in t6A tRNA modification due to GON7 and YRDC mutations lead to Galloway-Mowat syndrome" by Arrondel et al. presents a comprehensive study of two proteins involved in the t6A biosynthesis, namely YRDC and GON7, one of the five KEOPS proteins in humans. Mutations in these proteins can lead to the Galloway-Mowat syndrome (GAMOS), a rare neuro-renal disorder. Overall the manuscript is well-written and concise and shows in-depth in vivo and vitro data.

The results in the manuscript are essentially separated in two parts, a genetic/biochemical part and a structural investigation.

In the first part, through whole-exome sequencing of several patients with GAMOS, disease-related mutations in GON7 and YRDC were identified. Mutations in YRDC result in a more severe clinical outcome, suggesting that mutations in GON7 can be better compensated, as supported by cellular assays and the direct quantification of t6A content. In the second part, the authors seek to explain the in vivo/vitro findings for YRDC and GON7 on a structural biology level. They investigate the structural basis via homology models as well as NMR, SAXS and X-ray crystallography.

In the following, I will elaborate particularly on the second part, which requires some major revisions:

1) The in silico model of human YRDC must be treated with considerable caution as it relies on a crystal structure from yeast with only 20% sequence identity. Please add details on how the homology model was created.

This point has also been raised by reviewer 1 and our answers can be found below the remarks of reviewer 1 (first paragraph).

2) The conclusions based on the structural model for YRDC are not supported by experimental data, at best the in silico model might give some vague indications. Since YRDC was identified as the clinically more significant one compared to GON7, some experimental, structural data should be included for YRDC as well. Ideally, of course, solving the atomic resolution structure of human YRDC can allow for a more detailed understanding of the molecular mechanism behind the dramatic impact of various YRDC mutations on a clinical level. Alternatively, the folding/stability can be also probed by NMR spectroscopy, which was already employed in this study as a high-resolution technique. The NMR investigation of different YRDC mutants can be conclusive in terms of folding/stability, for example, when comparing 2D 1H,15N hetNOE, HSQCs/HMQCs or even 1D 1H spectra. Potentially, this would significantly strengthen the in silico model.

To strengthen the *in silico* YRDC model, we compared the stability and structure of the WT YRDC with those of the p.Ala84Val and p.Leu265del mutants. We expressed and purified the proteins in an *E. coli* expression system. The three proteins could be purified, but we noticed that the p.Ala84Val and p.Leu265del mutants were less stable and less soluble compared to the WT (6 mg/ml for WT, 3 mg/ml for p.Ala84Val and 1 mg/ml for p.Leu265del mutants) (see **Supplementary Methods section**). The samples precluded recording high-resolution structural NMR data. However, the solutions allowed recording of 1D H-NMR data (**Supplementary Fig.5a**). The spectra showed good dispersion of the N-H protons and of the methyl protons around the 0-1 ppm region suggesting the WT and mutants were well folded. These data however are not sufficient to conclude about any possible subtle structural differences between WT and mutants. We therefore decided to measure their catalytic activities *in vitro* (see **Supplementary Methods section**). The data are presented in the main text (lines 210-223) and the figure presenting the enzymatic activities is found in **Supplementary Fig.5b**. Interestingly, the two mutants displayed a clear loss in catalytic activities compared to WT, but retained sufficient activity to be able to complement YrdC (sua5) deletion in yeast.

3) The authors investigate the folding of human GON7 via NMR spectroscopy and, therefore, recorded 2D 1H,15N HMQC spectra. I agree, that the GON7 spectrum strongly suggests, that GON7 is unstructured based on the narrow chemical shift dispersion (SI Figure 4a, blue). But I do not understand the spectrum of GON7+LAGE3 (SI Figure 4a, red). The authors cautiously state, that "adding non-labelled LAGE3 to the

sample caused many chemical shift displacements, suggesting GON7 interacts with LAGE3". Comparing both 2D spectra reveals that peaks almost exclusively disappear, in fact, there are very few shifts. This is surprising. One would expect, that a molecular interaction between GON7 and LAGE3 necessarily implies (partial) folding of GON7. However, the GON7 spectra with and without LAGE3 look essentially the same, there are no additional peaks appearing in the spectral regions indicative of secondary structure elements. How do the authors explain this? How does the spectra look at low contour levels close to the noise level?

Are the peaks from folded regions simply clipped by the high contour threshold? The authors should also add the gel filtration profiles of the purified GON7+LAGE3 sample for the NMR study. I am stressing this as it would be a significant point to make, if the presence of LAGE3 induces in vitro folding of GON7. Or, the alternative, does partial folding require also the OSGEP protein? For the orthologous proteins in yeast, the dimeric complex of Gon7 and Pcc1 was sufficient to induce partial folding and allowed to determine the crystal structure. Is there a major mechanistic difference between human and yeast?

We observed by SAXS that LAGE3 and GON7 have a strong tendency to form a higher order complex (heterotetramer). We therefore suggest that the addition of LAGE3 to labeled GON7 causes the disappearance of a larger number of peaks in the HSQC spectrum due to slow tumbling of the complex. Those crosspeaks most likely correspond to residues directly involved in the interaction with LAGE3. Playing with the contour levels did not reveal the appearance of extra peaks. The peaks of the HSQC spectrum that are not affected probably correspond to the residues present in regions that remain unfolded in the GON7/LAGE3 complex and away from the interface. The crosspeaks that shifted in the spectra most likely correspond to residues that are close to the interface with LAGE3, but they are not necessarily directly involved into the interaction and still can retain significant mobility with respect of the large molecular weight complex.

We made this point explicit in the manuscript (lines 251-258 and lines 271-275)

We added the gel filtration profiles of the purified LAGE3/GON7, showing they form a homogenous complex (Supplementary Fig.13). The purification protocol of this complex was added to the Supplementary Methods section. These data were however not sufficient enough to conclude that OSGEP does not play a role in the folding of GON7 as well. We therefore added the P(r) function for the LAGE3/GON7 to the distance distribution curve shown in Supplementary Fig.6b and included data for the LAGE3/GON7 Kratky plot in main Fig.5a. These experiments highly suggest that the complex formation between LAGE3 and GON7 is sufficient to induce the folding of GON7. The Kratky plot of GON7 alone suggests that GON7 is greatly disordered. Adding LAGE3 induces a considerable compaction of GON7, confirming the crystallographic data. Further adding of OSGEP does not significantly change the shape of the Kratky plot, suggesting that it does not enhance folding of GON7 (Fig. 5a).

Similar observations were made for the Gon7/Pcc1 complex from yeast, where we could show that a heterotetramer is formed through dimerization of Pcc1 (Zhang et al., 2015).

4) Please provide the experimental X-ray scattering curves for GON7 and GON7+LAGE3. Is there a concentration dependence, which indicates aggregation? That could also explain the loss of signals in the NMR spectrum.

See reply to remark 3 [experimental X-ray scattering curves for GON7 and LAGE3/GON7 have been deposited to SASBDB (see reply to reviewer 2)].

There was no evidence for aggregation as judged from SAXS experiments at two different concentrations of GON7/LAGE3 (0.7 and 4.2 mg/ml). However, SAXS data show heterotetramer formation, explaining loss of NMR signals.

5) The authors present a high-resolution crystal structure of the GON7/LAGE3/OSGEP complex, which reveals a direct interaction of GON7 to LAGE3 but not with the catalytic OSGEP protein. Strikingly, the binding mode of human GON7/LAGE3 is structurally conserved with respect to the yeast Gon7/Pcc1 despite the very low sequence identity. With the structure at hand, can the authors bring the clinical impact of the GON7 mutations more into a structural context? How do mutations in GON7 impair the KEOPS complex? The manuscript still lacks a link between disease-related mutations and the complex structure. This should be made more clear as the structure is somewhat unattached.

The GAMOS c.21 C>A (pTyr7*) mutation in GON7 is a stop codon occurring very early in the nucleotide sequence, which likely leads to a complete absence of the protein. Sicheri's team demonstrated that the presence of Gon7 induces a 3-4 fold increase in catalytic activity of the KEOPS complex *in vitro*, suggesting the Gon7 is required for full expression of the t⁶A activity. Here we show that the deletion of GON7 drastically affects the stability of the other KEOPS partners in GON7 patient lymphoblastoid cell lines (Fig.6c). We can therefore conclude that GON7 is both necessary for stability and full activity of the KEOPS complex as explained in the discussion (lines 385-390).

6) Minor point: In the experimental details given in the Supplementary Information the authors describe in their expression protocol that they have used 0.5 μ M IPTG for induction. I believe this should read 0.5 mM IPTG.

This error was corrected

It is obvious that it is a major effort combining heterogeneous results as presented in this manuscript. The authors show some very interesting *in vivo* and *in vitro* findings, however, the structural biology part, which I can mainly assess, requires major revisions. It still lacks a more clear connection of the complex structure to the rest of the manuscript and further conclusions on how this structure can be helpful, e.g. in the context of medical treatment.

Our opinion on this particular matter diverges from that of the referee.

- 1) Our structure shows very nicely how the absence of the GON7 protein exposes a very hydrophobic surface that creates less active octamers *in vitro* and heavily destabilizes the KEOPS complex *in vivo*.
- 2) We also show YRDC mutants are still quite active and hence that their very debilitating effects are due to a relative small effect on activity, showing that the full activity of this enzyme is required in a healthy organism

Although a medical treatment is a far away goal for this very early onset developmental disorder, it is through a better knowledge of the pathophysiology of the disease that therapeutic options could be developed in the future.

Anyway, the structural and functional studies performed here were crucial to confirm the pathogenic role of the identified mutations, allowing in turn to provide the patient families with a precise diagnostic and offer them an adequate genetic counseling.

Reviewer #5 (Remarks to the Author):

In the manuscript "Defects in t⁶ A tRNA modification due to GON7 and YRDC mutations lead to Galloway Mowat syndrome", Arrondel et al describe 7 families with mutations in GON7 and YRDC leading to Galloway Mowat Syndrome, a severe form of syndromic nephrotic syndrome. The work is well presented and seems robust. Prior work of the some of the same authors on genes in the same KEOPS complex provides additional support to these findings.

Only few minor details will need to be addressed.

I would like the authors to add additional population allele frequencies for the likely pathogenic variants, since the families all come from areas of the world that were not well sampled in currently available databases. I am confident that the results presented here are robust, but it would be important to see population genetics data from ethnically matched controls, just to make sure that none of the variants reported here are observed in homozygous or compound heterozygous state in these populations.

We would like to thank the reviewer for this encouraging overall assessment. In our work, we identified two likely pathogenic variants (p.Ala84Val and p.Leu265del in the YRDC gene). These mutations were found in patients of European origin (Italy and the Netherlands, respectively), which is well represented in gnomAD. None of these mutations was present in this database. We acknowledge that it was not clearly indicated in our manuscript and therefore **edited Table 1** accordingly. We hope it adequately addresses the reviewer's comment.

Reviewers' Comments:

Reviewer #1:

Remarks to the Author:

The authors have satisfactorily responded to all of my questions and made the necessary changes to the manuscript. I do not have any further comments or suggestions. The paper certainly deserves publication.

Reviewer #2:

Remarks to the Author:

All my comments on the previous version have been addressed to my satisfaction. I recommend the manuscript for publication.

Reviewer #3:

Remarks to the Author:

The revised manuscript is improved over the original submission, and the authors responded to some of my previous concerns with new data and words. I remain concerned about the issue that GON7 mutation has almost no effect on t6A level (Fig 3d), but at least the nascent protein translation level decreased upon GON7 knockdown (Fig 4C), and I understand the differences between culture cells and tissues. The authors made concerted effort and discovered that GON7 mutation causes GAMOS, and revealed that mutant GON7 cannot sufficiently stabilize KEOPS complex, and made structural explanations. This manuscript reaches to the threshold of the publication in Nat Commun.

Reviewer #4:

Remarks to the Author:

The authors significantly improved the manuscript and addressed all inconsistencies raised with respect to the structural biology part of this study. The investigation of different YRDC mutants together with activity assays is convincing and seconds their structural model despite the low sequence identity. Also the addition of further NMR and SAXS experiments supports the here presented conception of an unfolded GON7 protein, which becomes partially folded in complex with LAGE3. The experiments also indicate that OSGEP is not vital for GON7 folding.

Minor revision

- Please add the amide part (~5-12 ppm) of the 1H 1D spectra.

Point-by-point response to reviewer's comments:

REVIEWERS' COMMENTS:

Only Reviewer #4 asked for one minor revision.

Reviewer #4 (Remarks to the Author):

The authors significantly improved the manuscript and addressed all inconsistencies raised with respect to the structural biology part of this study. The investigation of different YRDC mutants together with activity assays is convincing and seconds their structural model despite the low sequence identity. Also the addition of further NMR and SAXS experiments supports the here presented conception of an unfolded GON7 protein, which becomes partially folded in complex with LAGE3. The experiments also indicate that OSGEP is not vital for GON7 folding.

Minor revision

- Please add the amide part (~5-12 ppm) of the ¹H 1D spectra.

Supplementary Figure 5a now shows the amide part of the ¹H 1D spectra.